# Free-Bloom: Zero-Shot Text-to-Video Generator with LLM Director and LDM Animator

**Hanzhuo Huang**[*]  **Yufan Feng**[*]  **Cheng Shi**  **Lan Xu**  **Jingyi Yu**  **Sibei Yang**[†]

School of Information Science and Technology, ShanghaiTech University
https://github.com/SooLab/Free-Bloom

## Abstract

Text-to-video is a rapidly growing research area that aims to generate a semantic, identical, and temporal coherence sequence of frames that accurately align with the input text prompt. This study focuses on zero-shot text-to-video generation considering the data- and cost-efficient. To generate a semantic-coherent video, exhibiting a rich portrayal of temporal semantics such as the whole process of flower blooming rather than a set of "moving images", we propose a novel Free-Bloom pipeline that harnesses large language models (LLMs) as the director to generate a semantic-coherence prompt sequence, while pre-trained latent diffusion models (LDMs) as the animator to generate the high fidelity frames. Furthermore, to ensure temporal and identical coherence while maintaining semantic coherence, we propose a series of annotative modifications to adapting LDMs in the reverse process, including joint noise sampling, step-aware attention shift, and dual-path interpolation. Without any video data and training requirements, Free-Bloom generates vivid and high-quality videos, awe-inspiring in generating complex scenes with semantic meaningful frame sequences. In addition, Free-Bloom is naturally compatible with LDMs-based extensions.

## 1  Introduction

Recent impressive breakthroughs [31, 33, 35] in text-to-image synthesis have been attained by training diffusion models on large-scale multimodal datasets [37, 38] comprising billions of text-image pairs. The resulting images are unprecedentedly diverse and photo-realistic while maintaining coherence with the input text prompt. Nevertheless, extending this idea to text-to-video generation poses challenges as it requires substantial quantities of annotated text-video data and considerable computational resources. Instead, recent studies [53, 19, 40, 28, 5, 49, 17] introduce adapting pre-trained image diffusion models to the video domain in a data-efficient and cost-efficient manner, showing promising potential in one-shot video generation and zero-shot video editing.

This work takes the study of zero-shot text-to-video generation further, enabling the generation of diverse videos without the need for any video data. Instead of relying on a reference video for generation or editing [19, 28, 5, 40], the proposed approach generates videos from scratch solely conditioned on the text prompt. While a concurrent work, Text2Video-Zero [17], similarly focuses on zero-shot video generation, our study and Text2Video-Zero differ significantly not only in terms of technical contributions but also in motivation: we aim to generate complete videos that encompass meaningful temporal variations, which distinguish it from the generation with "moving images". As shown in Figure 1, with the text prompt of "a flower is gradually blooming", our approach generates a video that thoroughly depicts the entire process, seamlessly progressing from a flower bud to the

---

[*]Equal contribution.
[†]Corresponding author. yangsb@shanghaitech.edu.cn

37th Conference on Neural Information Processing Systems (NeurIPS 2023).

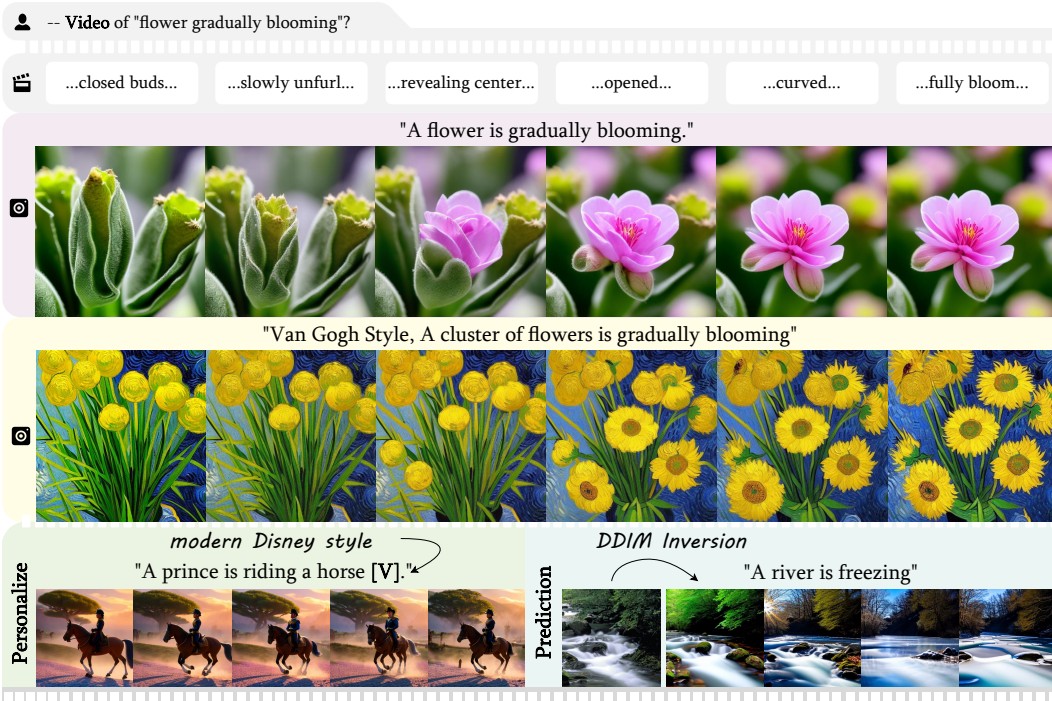

Figure 1: The zero-shot Free-Bloom generates vivid and high-quality videos that are imbued with semantic significance according to the input prompt. On top of this, Free-Bloom is compatible with the existing LDM-based extensions and can be applied to other tasks such as personalization of user-specific concepts and video prediction from a start frame.

full blooming stage. This set ours apart from other methods [17, 21, 10] that often portray a single stage of the flower bloom. In other words, our resulting video is semantic-coherent that exhibits a rich portrayal of temporal semantics.

In this work, we introduce Free-Bloom, a *zero-shot*, *training-free*, *semantic-coherent* text-to-video generator that is capable of generating high-quality, temporally consistent, and semantically aligned videos. Our key insight is to **harness large language models (LLMs) as the director, while pre-trained image diffusion models as the frame animator**. The underlying idea is that LLMs [7, 4, 30, 45], pre-trained on massive amounts of text data, encode general world knowledge [32, 27], thus being able to transform the text prompt into a sequence of prompts that narrates the progression of events over time. Simultaneously, pre-trained image diffusion models, such as latent diffusion models (LDMs) [33] generate the sequential frames conditioned on the prompt sequence to create a video. Moreover, considering the temporal resolution of the prompt sequence is difficult to fine-grained each frame of a video, thereby a zero-shot frame interpolation is employed to expand the video to a higher temporal resolution. Our insights shape Free-Bloom's progressive pipeline, which consists of three sequential modules: *serial prompting*, *video generation*, and *interpolation empowerment*.

The naïve attempt of directly applying LLMs and LDMs respectively to the first two steps of our pipeline resulted in failure. Not surprisingly, the resulting images are completely independent of each other. Although they each match with their own prompt, the semantic content of these images is entirely disjointed (*semantic coherence*), the overall content vary greatly including the foreground and the background (*identical coherence*), and the adjacent frames cannot be smoothly connected (*temporal coherence*). To overcome these problems, we propose a series of novel and effective technical solutions. *For semantic coherence*, we instruct LLMs in a multi-stage manner, completing the missing information in the generated prompt sequence caused by discourse cohesion, and ensuring that each prompt in the sequence accurately describes the detailed visual content while maintaining a consistent linguistic structure across prompts. *To ensure both identical coherence and temporal coherence*, we propose two innovative modifications for jointly adapting LDMs for video generation: (1) Through modeling the joint distribution of initial noise latent across frames from both unified and individual noise latent distribution, we enhance temporal and content consistency while preserving suitable levels of perturbation. Notably, in order to conform to the LDMs' assumption [33], our joint distribution ensures that the noise latent at every single frame follows normal Gaussian distribution

when the noise latent of other frames is not provided. (2) Considering the trade-off between continuity and adherence to single-frame semantics, we carefully incorporate cross-frame attention [53], which focuses on contextual frames, into self-attention layers according to denoising time steps.

We further propose the training-free interpolation empowerment module to improve the temporal resolution of the generated videos. In order to maintain the aforementioned semantic coherence, identical coherence, and temporal coherence, we consider semantic contents of both contextual and current frames.We perform a novel dual interpolation on latent variables relying on both contextual frames and self-denoising paths, as the former path enables smooth transitions between neighboring frames and the latter path ensures high fidelity of single frame.

In summary, our contributions are multi-fold: (1) We introduce Free-Bloom, a novel pipeline to tackle the zero-shot text-to-video task. Our pipeline effectively harnesses the rich world knowledge of LLMs and the powerful generative capability of LDMs, proposing an insight into adapting text-to-image models for video generation. (2) We propose a video generation module incorporating joint noise sampling and step-aware attention shift, ensuring identical coherence and temporal coherence while expressing the semantic content. (3) We introduce a training-free dual-path interpolation strategy, ensuring consistency with context while maintaining fidelity. (4) Free-Bloom can generate impressive videos imbued with semantic significance corresponding with the contextual narrative.

## 2  Related Work

**Diffusion Models for Images.** Diffusion models [42] and its variants DDPM [12] and DDIM [43], have achieved breakthroughs on text-to-image generation tasks [31, 8, 35, 33]. Moreover, diffusion models have a thriving research and application ecosystem, including multiple works [22, 11, 57, 23] as well as emerging open-source communities and libraries [48] with frameworks and plugins.

**Open-Domain Text-to-Video Generation.** Currently, both non-diffusion-based [15, 52, 15, 47] and diffusion-based [14, 41, 59, 21, 1, 13, 10, 3] T2V methods often conduct training on large-scale video datasets such as WebVid and HD-Vila-100M[2, 55], while leveraging T2I priors or jointly training with images to maintain the quality and the content diversity. However, even with datasets of millions of videos, the quality and quantity of the training set are still not comparable to images for training.

**Zero/Few-shot Video Synthesis.** Recently, tuning pre-trained T2I models under zero-shot/few-shot settings has also been found promising for video generation. Tune-A-Video [53] uses a reference video, generating videos conditioned on varied prompts while maintaining the original motion. Several works [19, 40, 28, 5, 49, 54] further explore the video editing task but are not able to either change the motion or generate videos with new events. Our concurrent work is Text2Video-Zero [17], which firstly proposes a zero-shot T2V pipeline started from pre-trained T2I models. While it models the motion flow by adding linear transformations to latent codes, we address that complex state transitions should be considered instead.

**LLM-assisted Generations.** Large Language Models (LLMs) [7, 4, 30, 45] have made significant impact by exhibiting remarkable performance across multiple Natural Language Processing tasks. From the immense training corpus, LLMs have captured open-world knowledge in various fields [24, 4, 27, 32, 44, 58]. To assist generation by LLMs, previous methods [51, 25, 20, 16, 39] are mainly based on prompt engineering, aiming to make the text prompts more expressive toward their goals. Our methods further exploit knowledge from LLMs to derive frame visual content in a complete video.

## 3  Preliminaries

**Denoising Diffusion Probabilistic Model.** DDPM [12] is a type of probabilistic model that learns to approximate the probability distribution of the true data. The forward diffusion process over $\mathbf{x}_0, \cdots, \mathbf{x}_T$ gradually adds Gaussian noises in $T$ time steps to corrupt an image. Then, the model is optimized to learn the denoising transitions $p_\theta(\mathbf{x}_{t-1}|\mathbf{x}_t)$ in the reverse process to turn noises into images. The forward posteriors can be expressed as

$$q\left(\mathbf{x}_{t-1} \mid \mathbf{x}_t, \mathbf{x}_0\right) = \mathcal{N}\left(\mathbf{x}_{t-1}; \tilde{\boldsymbol{\mu}}_t\left(\mathbf{x}_t, \mathbf{x}_0\right), \tilde{\beta}_t \mathbf{I}\right) \tag{1}$$

where $\tilde{\boldsymbol{\mu}}_t\left(\mathbf{x}_t, \mathbf{x}_0\right) := \frac{\sqrt{\bar{\alpha}_{t-1}}\beta_t}{1-\bar{\alpha}_t}\mathbf{x}_0 + \frac{\sqrt{\alpha_t}(1-\bar{\alpha}_{t-1})}{1-\bar{\alpha}_t}\mathbf{x}_t$ and $\tilde{\beta}_t := \frac{1-\bar{\alpha}_{t-1}}{1-\bar{\alpha}_t}\beta_t$.

**Denosing Diffusion Implicit Models.** DDIM [43] is a generalized form of DDPMs by introducing the non-Markovian processes. DDIM can speed up inference by using fewer denoising steps with the same training objective of DDPMs, where the reverse process can be written as follows,

$$\mathbf{x}_{t-1} = \sqrt{\alpha_{t-1}} \underbrace{\left( \frac{\mathbf{x}_t - \sqrt{1-\alpha_t}\boldsymbol{\epsilon}_t\left(\mathbf{x}_t;\theta\right)}{\sqrt{\alpha_t}} \right)}_{\text{``predicted } \mathbf{x}_0 \text{'' denoted as } \mathbf{P}_t} + \underbrace{\sqrt{1-\alpha_{t-1}-\sigma_t^2}\cdot\boldsymbol{\epsilon}_t\left(\mathbf{x}_t;\theta\right)}_{\text{``direction pointing to } \boldsymbol{x}_t \text{'' denoted as } \mathbf{D}_t} + \underbrace{\sigma_t\mathbf{z}_t}_{\text{random noise}} \quad (2)$$

where $\mathbf{P}_t := \frac{\mathbf{x}_t - \sqrt{1-\alpha_t}\boldsymbol{\epsilon}_t(\mathbf{x}_t)}{\sqrt{\alpha_t}}$ and $\mathbf{D}_t := \sqrt{1-\alpha_{t-1}-\sigma_t^2}\boldsymbol{\epsilon}_t(\mathbf{x}_t)$.

**Latent Diffusion Models.** LDMs [33] map data to a low-dimensional space, termed latent space, which possesses a strong representational capacity and can capture complex and abstract features. LDM employs a multi-layer attention-based U-Net architecture for noise prediction. The attention blocks [46] contain both self-attention and cross-attention, which focus on image content and textual content respectively. In the self-attention layer, the input feature $\mathbf{z}_i$ is projected into *query*, *key*, and *value* through linear transformations, then they are utilized to compute the attention weights.

## 4  Method

### 4.1  Free-Bloom: LLMs as The Director and LDMs as The Frame Animator

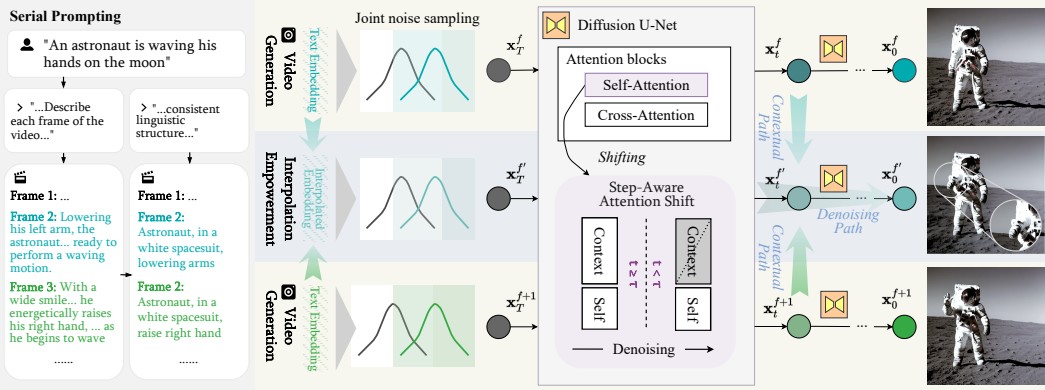

Figure 2: **Overview of Free-Bloom.** Our pipeline consists of three sequential stages. In *Serial Prompting* stage, the LLM is prompted to generate serial frame prompts. In *Video Generation* stage, modifications are made to the LDM to generate coherent frames by joint noise sampling and step-aware attention shift. In *Interpolation Empowerment* stage, a dual latent space interpolation conditioned on both contextual path and denoising path is proposed to generate intermediate frames.

We propose a zero-shot text-to-video generation pipeline named Free-Bloom, in which the generated videos are directed by LLMs and animated by LDMs. As shown in Figure 2, our pipeline consists of three sequential stages: *serial prompting*, *video generation*, and *interpolation empowerment*. Given a text prompt $\mathcal{T}$, Free-Bloom first generates a video $\mathcal{V}_1 \in \mathbb{R}^{f \times 3 \times h \times w}$ with low frame rates at the first two stages, which include $f$ frames with the frame size of $h \times w$. Then, the interpolation empowerment module fills in gaps between frames to improve continuity, resulting in the final video.

At the *serial prompting* stage, we first prompt the LLM with their general knowledge to transform the raw prompt into a series of prompts indicating the change of semantic content over time. Then, we instruct the LLM with referential resolution on the prompts to ensure that each prompt accurately describes the detailed visual content while maintaining a consistent linguistic structure across prompts, resulting in a prompt sequence, $\mathcal{T}^{1:f} = \{\mathcal{T}^1, \cdots, \mathcal{T}^f\}$. The prompt sequence accurately depicts the overall narrative and can effectively enlighten LDM [33] to generate semantic-coherence frames.

At the *video generation* stage, we employ two novel modifications to LDM, enabling it to generate semantic coherent, identical coherent, and temporal coherent video frames. We first simultaneously sample the noise at every frame from their joint Gaussian distribution, which is constructed by considering a unified noise at the video level and an individual noise at the frame level, which makes

it easier to generate coherence frames with certain variations. Then, we propose a strategy to modify the self-attention layer during the inference process. The modified attention layer adjusts attention to contextual information and self-consistent content according to the denoising time step.

At the *interpolation empowerment* stage, a dual latent space interpolation strategy is proposed to generate the intermediate frame between two neighboring frames. In addition to jointly interpolating the latent variables of neighboring frames for temporal coherence, we condition the latent variable generated by performing DDIM [43] on the interpolated text embedding to ensure semantic coherence. Also, the weights of the two paths vary over the denoising time step to ensure smoothly compatible.

## 4.2 Serial Prompting for Zero-Shot Text-to-Video Generation

**Prompting a Video.** When discussing the scenario of "a teddy bear jumping into the water", humans naturally imagine the series of events: the teddy bear crouches down, then propels itself into the air, and finally lands in the water with a splash. The prior knowledge of jumping water allows us humans to derive a semantic meaning of sequential events from a general description, which inspires us to ask: do LLMs also encode this kind of knowledge without further training? In our investigation of LLMs, we prompt the LLMs with instructions such as "describe each frame". We find that LLMs incorporate extensive world knowledge and can provide temporal transition knowledge.

However, the generated initial descriptions have free linguistic structure, and the text is fragmented in every single description due to discourse coherence, leading to falling short of adequately sufficient information for a single frame, as shown in Figure 2. Therefore, to ensure semantic coherence of prompts across frames, we further instruct the LLMs with the above-obtained initial descriptions to generate a sequence of $f$ prompts $\mathcal{T}^{1:f} = \{\mathcal{T}^1, \cdots, \mathcal{T}^f\}$ with consistent linguistic structures, where each prompt accurately describes the visual content in detail.

**Zero-Shot Video Generation with LDMs.** The video generation module aims to condition the LDMs on the prompt sequence $\mathcal{T}^{1:f}$ to generate frames that are *semantic, identical, and temporal coherence*. However, simply generating $f$ still images from different prompts results in a collection of unrelated images that cannot be sequenced into a coherent video. To address this issue, we propose two novel modifications for adapting LDMs to generate videos rather than a collection of images without the need for additional training.

• *Joint noise sampling following LDMs' assumption.* To model the coherence across frames, we propose to sample the initial noises in the diffusion process of frames from their joint probability distribution instead of independent distribution. To construct such joint distribution, we first investigate the effects of independent noise and united noise over sequential frames: (1) The same noise for every frame results in LDMs generating a sequence of images with similar content under similar textual conditions. While this feature may contribute to achieving temporal coherence among frames, it can potentially restrict the natural variation of the subject, significantly reducing the overall video quality. (2) Generating images from $f$ independent noises issues in maintaining consistency across frames, although the generated images with naturally varying content showcase more diversity and independently captivating elements.

These observations motivate us to construct the joint distribution by considering the independent and united distribution jointly. Specifically, we propose to obtain the initial noisy latent variable by weighted summation over a unified noise across the video frames and an individual perturbed noise to maintain consistency across frames while introducing appropriate variation. Moreover, to conform to the LDMs' assumption, we design the weighting coefficient to ensure that without giving initial noise latent at other frames, the initial noise at every single frame follows normal Gaussian distribution. Let us denote the unified video noise as $\mathbf{x}_T^*$ and the independent noise as $\mathbf{x}_T^i$, the joint distributions $\mathbf{x}_T^{1:f}$ and $\boldsymbol{\delta}_T^{1:f}$ are formulated as follows,

$$
\begin{aligned}
\mathbf{x}_T^{1:f} = [\mathbf{x}_T^*, \cdots, \mathbf{x}_T^*]^T, \mathbf{x}_T^* \sim \mathcal{N}(\mathbf{0}, \mathbf{I}_n) &\quad \Rightarrow \quad p(\mathbf{x}_T^{1:f}) = \mathcal{N}(\mathbf{0}, \mathbf{J}_f \otimes \mathbf{I}_n) \\
\boldsymbol{\delta}_T^{1:f} = [\mathbf{x}_T^1, \cdots, \mathbf{x}_T^f]^T, \mathbf{x}_T^i \sim \mathcal{N}(\mathbf{0}, \mathbf{I}_n) &\quad \Rightarrow \quad p(\boldsymbol{\delta}_T^{1:f}) = \mathcal{N}(\mathbf{0}, \mathbf{I}_{nf})
\end{aligned}
\tag{3}
$$

where $\mathbf{J}_f$ denotes the all-ones matrix with size of $f \times f$ and $\otimes$ represents the Kronecker product. Then, we model the mixture noise by

$$
\tilde{\mathbf{x}}_T^{1:f} := \cos(\frac{\pi}{2}\lambda)\mathbf{x}_T^{1:f} + \sin(\frac{\pi}{2}\lambda)\boldsymbol{\delta}_T^{1:f}
\tag{4}
$$

where the $\lambda$ is a coefficient of variance rate. This mixed noise follows the joint distribution as

$$p(\tilde{\mathbf{x}}_T^{1:f}) = \mathcal{N}(\mathbf{0}, \sin^2(\frac{\pi}{2}\lambda)\mathbf{I}_{nf} + \cos^2(\frac{\pi}{2}\lambda)\mathbf{J}_f \otimes \mathbf{I}_n))$$
$$= \mathcal{N}(\mathbf{0}, \mathbf{I}_{nf} + \cos^2(\frac{\pi}{2}\lambda)((\mathbf{J}_f - \mathbf{I}_f) \otimes \mathbf{I}_n)). \tag{5}$$

When $\lambda = 0$, the initial noise becomes unified noise, and when $\lambda = 1$, it becomes independent noise.

• *Step-aware attention shift.* To further maintain identical coherence while ensuring semantic coherence, we shift the attention of the self-attention layers in the LDMs' U-Net from cross-frame contexts to the current frame according to the denoising time steps. For frame $i$ with its *query* $Q_i$, previous methods [53, 5, 49, 28] with one overall text prompt retrieve the *key* and *value* from the former frame and the first frame to perform sparse spatio-temporal attention, based on the observation [53] that extending the spatial self-attention across images produces consistent content.

In our scheme, frames are conditioned on different prompts, requiring that not only maintaining temporal and identical coherence, but also semantic coherence with their respective prompts. *To achieve identical and temporal coherence*, we address the former and the first frame as *contextual* frames and attend to contextual key-value pairs. In particular, the former frame helps to enhance temporal coherence, while the first frame acts as a benchmark shape, maintaining appearance consistency throughout the video. *To align with the respective prompt in semantics*, we shift attention to the current frame itself as the time step increases, enabling the preservation of its specific characteristics and details. *In summary*, our inference strategy takes into account the time step during the diffusion process. The initial steps focus on contextual frames to form coarse-grained shapes and layouts. As we progress, complete images are generated, emphasizing producing accurate outlines conditioned on semantic information. Overall, our adaptation of the attention mechanism refers to:

$$\text{Self-Attention}_i := \begin{cases} \text{Attention}(Q_i, [K_0, K_{i-1}, K_i], [V_0, V_{i-1}, V_i]), & t \geq \tau \\ \text{Attention}(Q_i, K_i, V_i), & t < \tau \end{cases} \tag{6}$$

where $\tau$ is the threshold of the time step for attention shift. The $(K_0, V_0)$, $(K_{i-1}, V_{i-1})$, and $(K_i, V_i)$ are the key-value pairs in the first, former, and current frames, respectively.

### 4.3 Interpolation Empowerment

The interpolation empowerment module is proposed to further increase the frame rate without extra training resources. Similar to the insight of utilizing contextual information in the *"Step-aware attention shift"* proposed in Section 4.2, the interpolated intermediate frames should also consider contextual frames, *i.e.*, the former and the latter frames, in the denoising process. One naïve approach is to directly derive the latent variable of the intermediate frame solely from the latent variables in contextual frames. However, this method overlooks the intermediate semantics indicated in text prompts in contextual frames, leading the semantic incoherence.

Therefore, we propose *a dual interpolation path* for generating intermediate latent variables, where the contextual path interpolates the latent variables between the contextual frames to ensure temporal coherence, while the denoising path interpolates latent variables in DDIM [43] denoising process conditioned on interpolated text embedding to improve the semantic coherence. Specifically, to interpolate a new frame $\mathbf{x}^f$ between $\mathbf{x}^{f-1}$ and $\mathbf{x}^{f+1}$, we first sample the intermediate initial latent variable from the same distribution proposed in Section 4.2. For the conditional textual prompt $\mathcal{T}^f$, we directly interpolate text embeddings of the previous and next frames.

1) *Contextual path.* To obtain the context-sensitive latent variable $\tilde{\mathbf{x}}_t^f$, we perform linear interpolation between the contextual frames $\mathbf{x}_t^{f-1}$ and $\mathbf{x}_t^{f+1}$ as follows,

$$\tilde{\mathbf{x}}_t^f = k\mathbf{x}_t^{f-1} + (1-k)\mathbf{x}_t^{f+1}. \tag{7}$$

2) *Denoising path.* Then, we perform a linear interpolation between the context-sensitive latent variable $\tilde{\mathbf{x}}_t^f$ and the latent variable obtained from DDIM conditioned on the interpolated prompt $\mathcal{T}^f$. We employ $m(t)$ as the interpolation coefficient and use the notation of $\mathbf{P}_{t+1}^f$ and $\mathbf{D}_{t+1}^f$ mentioned in Section 3. The interpolation coefficient $m(t)$ varies according to the time step, with smaller values in the earlier denoising steps and increase in the latter steps. The interpolation is formulated as follows,

$$\mathbf{x}_t^f = (1 - m(t))\tilde{\mathbf{x}}_t^f + m(t)(\sqrt{\alpha_t}\mathbf{P}_{t+1}^f + \mathbf{D}_{t+1}^f + \sigma_t\mathbf{z}_t). \tag{8}$$

To provide an intuitive conditional probability distribution of $x_t^f$, we present the distributions for the cases when $m(t) = 1$ and $m(t) = 0$ respectively as:

$$p(\mathbf{x}_t^f | \mathbf{x}_{t+1}^f) = \begin{cases} \mathcal{N}(k\boldsymbol{\mu}_{t+1}^{f-1} + (1-k)\boldsymbol{\mu}_{t+1}^{f+1}, [k \quad 1-k]\, \boldsymbol{\Sigma}_{t+1}^{f-1,f+1} \begin{bmatrix} k \\ 1-k \end{bmatrix}), & m = 0 \\ \mathcal{N}(\boldsymbol{\mu}_{t+1}^f, \boldsymbol{\Sigma}_{t+1}^f), & m = 1 \end{cases} \quad (9)$$

where $\boldsymbol{\Sigma}_{t+1}^{f-1,f+1}$ is the covariance matrix of $\mathbf{x}_{t+1}^{f-1}$ and $\mathbf{x}_{t+1}^{f+1}$.

## 4.4 Extensions

Our method exhibits high extensibility and seamless integration with existing LDM-based methods. We demonstrate the versatility of our approach through some applications: (1) **Personalization.** By combining with LDM-based personalization methods [9, 34], our approach can generate videos with user-provided concepts. Here we choose DreamBooth [34] as an example, as shown in the top Figure 1, our method generates a video of "A prince is riding a horse [V]" with the concept [V] as "modern Disney style". (2) **Making an image move.** With the help of DDIM inversion [43], we can inverse the latent variable from an image, then generate a sequence of frames that continues the image. For example, starting from a still image of the scenery of a river, our method further generates a video depicting the process of freezing, as illustrated in Figure 1.

## 5 Experiments

**Implementation Settings.** We develop our Free-Bloom based on LLM as ChatGPT [26] and LDM as Stable Diffusion [33] with its pre-trained v1.5 weights. We first generate a video of $f = 6$ length with $512 \times 512$ resolution, then we iteratively interpolate between the most distinguished frames with $k = 0.5$ for the first interpolated frame. For $m(t)$, we set $m(t) = 0.1$ when the denoising time $t \geq \tau^*$ and $m(t) = 1$ when $t < \tau^*$. Notice that our method actually can be adapted to generate longer videos. To enhance image quality in practice, we add fixed negative prompts and upscale our resulting frames with an image super-resolution network ESRGAN [50]. All experiments are performed on a single NVIDIA GeForce RTX 3090Ti.

### 5.1 Baseline Comparisons

**Qualitative Results.** We showcase examples of Free-Bloom in Figure 3 and compare them with that generated by the zero-shot T2V-Zero [17] and VideoFusion [21], which demonstrate the most outstanding overall performance in the user study. More comparisons are included in the Supplementary. Video generation shown in region A demonstrates the following observations: (1) Our method vividly depicts the complete imagery of a volcanic eruption or the sequential motion of a teddy bear jumping into the water, exhibiting the capacity to generate semantic meaningful frame sequences, in which the visual elements, actions, and events are aligned with the input prompt as well as the contextual narrative. (2) In addition, our method shows temporal coherence and identical coherence while maintaining high fidelity for single frames. (3) Although the T2V-Zero method maintains overall content consistency between frames, it fails to depict sequential events. Furthermore, the subject would easily be distorted as the length of the video increases. (4) VideoFusion, on the other hand, demonstrates impressive temporal coherency between frames as it is trained on large-scale datasets, and it also presents a certain level of grasp of events. However, this training on the vast video dataset also significantly degrades the fidelity and quality of individual frames.

For interpolation results shown in region B, we present one latest state-of-art Video Frame Interpolation method AMT [18] pre-trained on Vimeo90K [56] for comparison. AMT fails to comprehend the subject information between the two target frames. As a result, it blurs the different parts of the teddy bear's body, failing to capture the intermediate motion. In contrast, our method fills in the content gap with a serial of continuous motion of the bear from the air to the water, maintaining fidelity in the intermediate frames while ensuring content consistency.

**Quantitative Results** are reported with automatic metrics and the user study in Table 1. We adopt three publicly available diffusion-based methods, Text2Video-Zero [17], VideoFusion [21], and LVDM [10] as baselines. VideoFusion and LVDM are both trained methods while the former is

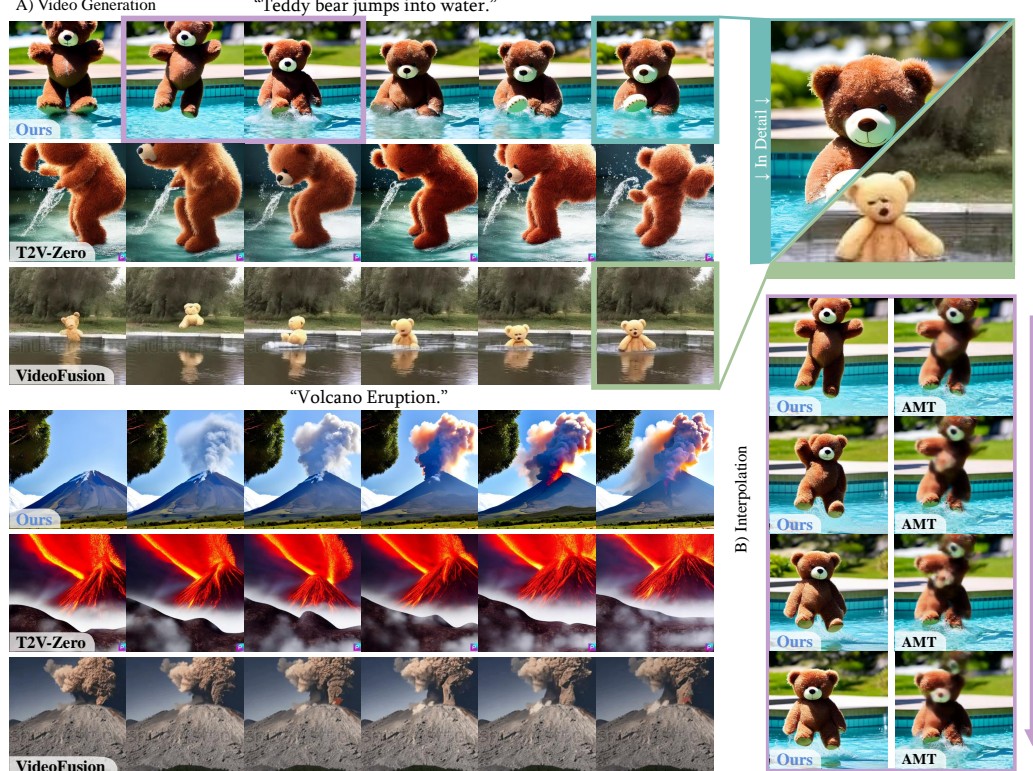

Figure 3: **Qualitative comparison.** A) Our method can generate semantic meaningful frame sequences when conditioned on the same prompts. B) We interpolate two frames into the frame sequence by taking the two images within the same color box in part A as the start and the end.

trained on both large-scale image datasets ImageNet [6], LAION5B [38] and a large-scale video dataset WebVid-10M [2], and the latter has been trained on a 2-million subset of the WebVid-10M.

For automatic metrics, we use CLIP [29] to evaluate the similarity correlation between the input prompts and the visual content of generated frames. Recall that our method put an emphasis on the overall narrative semantic coherence, therefore we also compute the CLIP score between each frame and its corresponding prompt generated by LLM(*). We can observe that although comparing each frame against the input prompt overlooks the potential of our approach, our method demonstrates good performance when frames are compared with the frame-level video prompts.

For the user study, the participants are instructed to rate the fidelity, temporal coherence, and semantic coherence on a scale of 1 to 5 and give a comprehensive ranking. According to the user study, although our method may not perform as well as trained methods in terms of temporal continuity, it has received high recognition in all other dimensions of video quality.

Table 1: **Quantitative Results.** * for CLIP score on serial prompts.

| Method | Training-Free | Automatic Metric | User Study | | | |
|---|---|---|---|---|---|---|
| | | CLIP Metrics↑ | Fidelity ↑ | Temporal ↑ | Semantic ↑ | Rank ↓ |
| VideoFusion [21] | | 0.483 | 3.436 | 3.889 | 3.267 | 2.317 |
| LVDM [10] | | 0.480 | 3.289 | 3.650 | 3.242 | 2.567 |
| T2V-Zero [17] | ✓ | 0.479 | 3.486 | 2.783 | 3.025 | 3.033 |
| Ours | ✓ | 0.477 / 0.482* | 4.133 | 3.267 | 3.867 | 2.083 |

## 5.2 Analysis of Our Pipeline

In Figure 4 and Figure 5, we conduct a comprehensive analysis of the modules in our proposed pipeline. The top row showcases our final results.

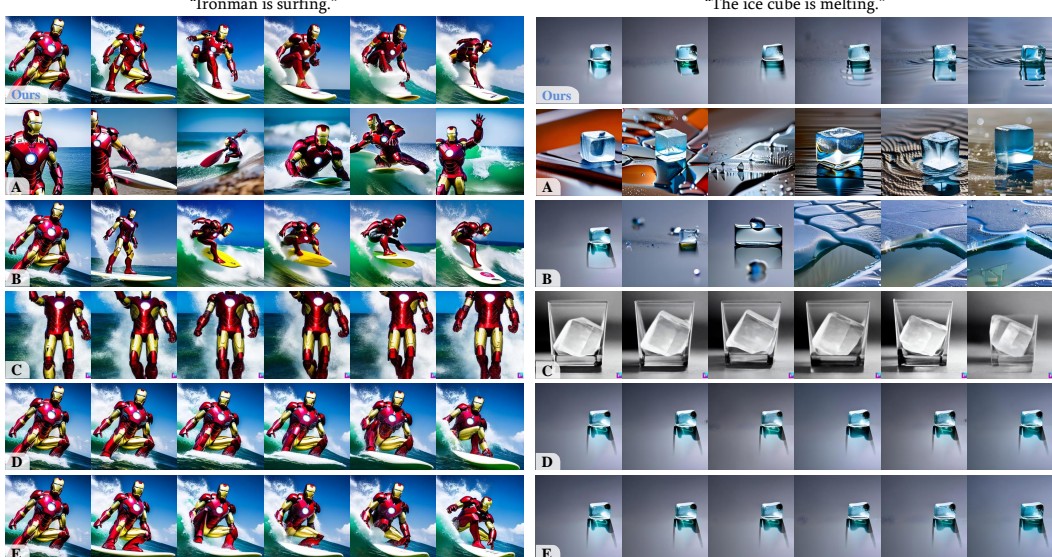

"Ironman is surfing."        "The ice cube is melting."

Figure 4: Analyze for the effects of Free-Bloom (A) Without joint noise sampling, (B) Without shifting self-attention, (C) Serial Prompting for Text2Video-Zero, (D) Without attention to the current frame itself, and (E) Without the step-aware shift strategy.

*A) Without joint noise sampling.* In A, we replace the joint noise initialization with the independent initialization, resulting in frame sequences with inconsistent image content. *B) Without shifting self-attention.* In B, we denoise the frames from the proposed joint noise initialization but use the raw LDM without modifying self-attention layers. The frames are similar naturally in some cases, as the joint noise contains a portion of unified noise. However, without attention to contextual frames, it is difficult to maintain identical coherence of both the foreground and background content, let alone temporal continuity. *C) Serial Prompting for Text2Video-Zero.* In C, we adapt Text2Video-Zero [17] to enable the input of frame-level video prompts so that each frame is generated and conditioned on a distinct prompt. The results show that it is challenging for the current method to comprehend serial prompts effectively, resulting in "moving images". *D) Without attention to the current frame itself.* In D, we replace all self-attention layers with spatio-temporal attention layers proposed in TAV [53]. The resulting frames exhibit improved temporal coherence, demonstrating smoother transitions between frames. However, the frames appear almost identical, creating a sequence of meaningless and jittery frames based solely on the first frame, which does not align with the intended temporal semantics. Additionally, prolonged contextual attention in long sequences can significantly compromise the fidelity of individual frames. As shown in the case of Iron Man, the last frame presents an incomplete leg. *E) Without the step-aware shift strategy.* Based on D, we further concatenate the attention to the current frame with contextual attention but without shifting it along time steps. Although the results remain inconsistent with semantics due to the absence of the inference strategy that varies across the denoising time step, the fidelity of images is improved.

**Dual-path interpolation.** We show the influence of balancing timing coefficients between the dual interpolation path in Figure 5. When $\tau^*$ approaches 1, $m(t)$ remains relatively small throughout the entire denoising process, indicating that the contextual path primarily determines the interpolated results. The resulting image presents

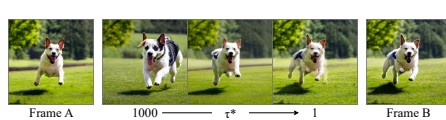

Frame A     1000 ——— $\tau^*$ ——→ 1     Frame B

Figure 5: Analyze for interpolation.

blurring and ghosting artifacts, failing to generate a frame with correct semantics. Conversely, heavily relying on the DDIM denoising path with larger values of $m(t)$ causes substantial deviations in content compared to the contextual frames. Our method adopts smaller values of $m(t)$ in the early steps to prioritize coarse-grained shapes and layout and increases $m(t)$ in the latter steps to improve temporal consistency. As a result, we achieve the benefits of both paths by appropriately setting $m(t)$.

# 6    Conclusion

In this paper, we devise a novel zero-shot and training-free text-to-video approach, which mainly focuses on improving the narrative of the progression of events. Our proposed pipeline effectively harnesses the knowledge from the pre-trained LLM and LDM and produces highly semantic coherent videos while also maintaining temporal coherence and identical coherence. **Limitation**: we look forward to further research on text-to-video generation, however, it should be acknowledged that there can be ethical impacts like other generative models. As we adopt ChatGPT and Stable Diffusion v1.5, our method may inherit the bias of those two models.

**Acknowledgment:** This work was supported by the National Natural Science Foundation of China (No.62206174), Shanghai Pujiang Program (No.21PJ1410900), Shanghai Frontiers Science Center of Human-centered Artificial Intelligence (ShangHAI), MoE Key Laboratory of Intelligent Perception and Human-Machine Collaboration (ShanghaiTech University), and Shanghai Engineering Research Center of Intelligent Vision and Imaging. Also, we extend our heartfelt gratitude to the anonymous users who participated in our user study.

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

# Appendix

In this section, we provide additional discussions, details, and experiments to further support our contributions. The content is organized as

## A    Discussions

**Limitations and Future Work.** We look forward to further research on this method. While our method offers the advantage of being training-free and not requiring extra training data, it highly depends on the large foundation models LLMs [7, 4, 26] and LDMs [33]. Consequently, it would inherit the limitations of those large pre-trained models. For example, LDMs often struggle with generating images containing detailed faces and limbs, specific text, multiple objects, interactions between objects, etc, therefore our method has the same weakness. Moreover, LDMs are often sensitive to seed selections of initial noises [36], so when the initial frame is of low quality, our method tends to result in relatively poor performance as well. Additionally, although our method demonstrates improved temporal consistency to other zero-shot methods, we found that it is still challenging to maintain high temporal coherency between frames in the zero-shot setting. However, leveraging video data proves to be an effective solution for acquiring temporal priors. Therefore, how to combine the strengths of zero-shot methods and trained methods is a promising direction for future research.

**Societal Impacts.** It should be acknowledged that there can be ethical impacts like other generative models. As we adopt ChatGPT [26] and Stable Diffusion v1.5 [33], our method may inherit the bias of those two models. Also, although the results of our method of direct text-to-video generation are still a step away from convincingly photo-realistic videos, the risk of abuse, for example, generating fake, harmful, or discriminating content, should be aware.

## B    Joint Noise Derivation

First, let us consider the distribution of **unified noise**. It is composed by initial noise $\mathbf{x}_T^* \sim \mathcal{N}(\mathbf{0}, \mathbf{I}_n)$ for each frame and can be represented as $\mathbf{x}_T^{1:f} = [\mathbf{x}_T^*, \cdots, \mathbf{x}_T^*]^T$. The noise of any two frames in $\mathbf{x}_T^{1:f}$ have the same values, and therefore their covariance is

$$\mathrm{cov}(\mathbf{x}_T^*, \mathbf{x}_T^*) = \mathbf{D}(\mathbf{x}_T^*) = \mathbf{I}_n. \tag{10}$$

Thus the unified noise follows the distribution as

$$p(\mathbf{x}_T^{1:f}) = \mathcal{N}(\mathbf{0}, \mathbf{J}_f \otimes \mathbf{I}_n), \tag{11}$$

where $\mathbf{J}_f$ represents the all-one matrix of size $f \times f$ and $\otimes$ denotes the Kronecker product. The specific form of $\mathbf{J}_f \otimes \mathbf{I}_n$ is

$$\mathbf{J}_f \otimes \mathbf{I}_n = \overbrace{\begin{bmatrix} \mathbf{I}_n & \cdots\cdots & \mathbf{I}_n \\ \vdots & \ddots & \vdots \\ \mathbf{I}_n & \cdots\cdots & \mathbf{I}_n \end{bmatrix}}^{f \times \mathbf{I}_n} \Bigg\} f \times \mathbf{I}_n \tag{12}$$

Second, let us consider the distribution of **individual noise**, in which each frame is independently sampled. Therefore, the covariance between any two frames is $\mathbf{0}$, and the distribution still follows a standard normal distribution:

$$p(\boldsymbol{\delta}_T^{1:f}) = \mathcal{N}(\mathbf{0}, \mathbf{I}_{nf}) \tag{13}$$

According to Section 4.2, the mixed noise is defined as $\tilde{\mathbf{x}}_T^{1:f} := \cos(\frac{\pi}{2}\lambda)\mathbf{x}_T^{1:f} + \sin(\frac{\pi}{2}\lambda)\boldsymbol{\delta}_T^{1:f}$. Since $\mathbf{x}_T^{1:f}$ and $\boldsymbol{\delta}_T^{1:f}$ are independently sampled, the sum of the two still follows a normal distribution, with a mean of $\mathbf{0}$ and a variance of

$$
\begin{aligned}
\sin^2(\frac{\pi}{2}\lambda)\mathbf{I}_{nf} + \cos^2(\frac{\pi}{2}\lambda)\mathbf{J}_f \otimes \mathbf{I}_n) &= \sin^2(\frac{\pi}{2}\lambda)\begin{bmatrix} \mathbf{I}_n & \mathbf{0} & \cdots\cdots & \mathbf{0} \\ \mathbf{0} & \ddots & \ddots & \vdots \\ \vdots & \ddots & \ddots & \mathbf{0} \\ \mathbf{0} & \cdots\cdots & \mathbf{0} & \mathbf{I}_n \end{bmatrix} + \cos^2(\frac{\pi}{2}\lambda)\begin{bmatrix} \mathbf{I}_n & \cdots\cdots & \mathbf{I}_n \\ \vdots & \ddots & \vdots \\ \mathbf{I}_n & \cdots\cdots & \mathbf{I}_n \end{bmatrix} \\
&= \begin{bmatrix} \mathbf{I}_n & \cos^2(\frac{\pi}{2}\lambda)\mathbf{I}_n & \cdots & \cos^2(\frac{\pi}{2}\lambda)\mathbf{I}_n \\ \cos^2(\frac{\pi}{2}\lambda)\mathbf{I}_n & \ddots & & \cos^2(\frac{\pi}{2}\lambda)\mathbf{I}_n \\ \vdots & & \ddots & \vdots \\ \cos^2(\frac{\pi}{2}\lambda)\mathbf{I}_n & \cos^2(\frac{\pi}{2}\lambda)\mathbf{I}_n & \cdots & \mathbf{I}_n \end{bmatrix} \\
&= \mathbf{I}_{nf} + \cos^2(\frac{\pi}{2}\lambda)((\mathbf{J}_f - \mathbf{I}_f) \otimes \mathbf{I}_n)
\end{aligned}
\tag{14}
$$

Thus, variable $\tilde{\mathbf{x}}_T^{1:f}$ follows the following distribution.

$$
\begin{aligned}
p(\tilde{\mathbf{x}}_T^{1:f}) &= \mathcal{N}(\mathbf{0}, \sin^2(\frac{\pi}{2}\lambda)\mathbf{I}_{nf} + \cos^2(\frac{\pi}{2}\lambda)\mathbf{J}_f \otimes \mathbf{I}_n)) \\
&= \mathcal{N}(\mathbf{0}, \mathbf{I}_{nf} + \cos^2(\frac{\pi}{2}\lambda)((\mathbf{J}_f - \mathbf{I}_f) \otimes \mathbf{I}_n))
\end{aligned}
\tag{15}
$$

In this distribution, without given noises of other frames, for any frame noise $\tilde{\mathbf{x}}_T^i$, it still follows a standard normal distribution that $p(\tilde{\mathbf{x}}_T^i) = \mathcal{N}(\mathbf{0}, \mathbf{I}_n)$.

## C  Implementation Details

### C.1  Serial Prompting

We first prompt ChatGPT [26] with the following instruction:

- *I would like you to play the role of the describer of each frame of the video as a director of a movie. The content of each video should be concise and only clearly describe the subject. Each sentence in the video is independent. Every sentence needs to include the subject's appearance and actions, please describe the main actions of the object and the extent of the actions in as much detail as possible. The sentences of each picture are independent, and each sentence should describe what exists in the picture. Each frame is described in only one sentence. Suppose there is a video about "[INPUT PROMPT]" and there are "[F]" frames in the video. Describe the content of each frame separately. Please be straightforward and do not use a narrative style.*

Then, we use the following prompt:

- *Now perform Coreference Resolution on the above sentence, replace reflexive pronouns with their original vocabulary, and eliminate the discourse cohesion. Keep the meaning the same. The sentence for each frame should be able to fully express all the visual information of the frame. Also, the linguistic structure of each sentence should be simple and similar.*

## C.2 Test Set for Quantitative Results

We list some prompts from our test set in Table 2, in which some prompts are from the webpage of Text2Video-Zero [17] and some are designed by ourselves that incorporate more complex event content.

Table 2: Prompts for Test Set.

| | |
|---|---|
| A cluster of flowers blooms | Astronaut riding a horse |
| Use pan to fire an egg | Iron man is surfing |
| Volcano eruption | The ice cube is melting |
| A dog is walking down the street | A panda is walking down the street |
| Light a match then the match goes out | The Santa flying through the sky |
| River freezes | Two supermen are fighting |
| Two men shake hands | A bear dancing on times square |
| Teddy bear jumps into water | An astronaut is waving his hands on the moon |
| The growth of a sapling | An egg hatch into a chick |
| A dancing mickey | Teddy bear is greeting |

## C.3 Details of User Study

We conduct a user study to understand how humans would evaluate the current text-to-video methods. The survey contains a total of 20 prompts with each prompt having 4 videos output from VideoFusion [21], LVDM [10], Text2Video-Zero [17], and Ours. For each prompt, we ask raters to answer the following four questions:

 • How would you rate the temporal coherence and smoothness of the videos? Please assign a score for their **continuity**. (*Temporal Coherence*)

 • How would you rate the quality and fidelity of the individual frames in the videos? Please assign a score for the **visual quality**. (*Fidelity*)

 • How well does the video depict the content described in the text? Please assign a score for its **content** representation. (*Semantic Coherence*)

 • Based on your **overall perception**, please rank the videos. (*Rank*)

Figure 14 presents an example interface of our survey. We received valid responses from a total of 80 individuals from both industry and academia.

## C.4 Code Used and License

Table 3: The used codes and license.

| URL | Citation | License |
|---|---|---|
| https://github.com/showlab/Tune-A-Video | [53] | Apache License 2.0 |
| https://github.com/google/prompt-to-prompt | [11] | Apache License 2.0 |
| https://github.com/huggingface/diffusers | [48] | Apache License 2.0 |
| https://github.com/Picsart-AI-Research/Text2Video-Zero | [17] | CreativeML Open RAIL-M |
| https://github.com/VideoCrafter/VideoCrafter | [10] | (Hugging Face Space) MIT |
| https://github.com/modelscope/modelscope/ | [21] | Apache License 2.0 |

All used codes and their licenses are listed in Table 3.

# D Additional Experiments

## D.1 Qualitative Results

We showcase more visualization of the generated video in this section. In Figure 6 and Figure 7, we present the full comparison with Text2Video-Zero [17], VideoFusion [21], and LVDM [10]. In Figure 8, we randomly generate multiple videos with respect to the same prompts. In Figure 9, we demonstrate more results of our interpolation empowerment module.

## D.2 User Study Quantitative Comparisons

Table 4: **User Study Comparisons.**

| Method | Training-Free | User Study | | | |
|---|---|---|---|---|---|
| | | Rank | Fidelity | Temporal | Semantic |
| Ours vs. LVDM [10] | | 55.00% | 85.28% | 53.33% | 82.27% |
| Ours vs. VideoFusion [21] | | 63.06% | 82.50% | 44.44% | 78.33% |
| Ours vs. T2V-Zero [17] | ✓ | 73.61% | 87.78% | 80.00% | 85.28% |

For the user study part in the quantitative results, we also present the comparison-based results here in Table 4. Specifically, for the ranking column, the number denotes the percentage of participants who prefer our method and rank us before another. For the dimensions of fidelity, temporal coherence, and semantic coherence, the numbers indicate the percentage of participants who believe that our generated videos are better to that of another method in that dimension.

## D.3 Analysis on Joint Noise Sampling

We additionally analyze the effect of our noise sampling method. In part A of Figure 10, we sample the noise the same as equation $\tilde{\mathbf{x}}_T^{1:f} := \cos(\frac{\pi}{2}\lambda)\mathbf{x}_T^{1:f} + \sin(\frac{\pi}{2}\lambda)\boldsymbol{\delta}_T^{1:f}$ using $\sin\cos$ weighting. While in part B, we modify the weight of the unified noise and the individual noise as

$$\tilde{\mathbf{x}}_T^{1:f} := (1 - \lambda)\mathbf{x}_T^{1:f} + \lambda\boldsymbol{\delta}_T^{1:f} \tag{16}$$

However, this would disrupt the single-frame noise from following normal Gaussian Distribution. As we can observe, in this way, LDM fails to generate reasonable images.

## D.4 Extensions

In this section, we demonstrate the extensibility of our approach with more examples. Inspired by Phenaki [47] which can generate story-based conditional videos based on a sequence of prompts, we also apply our method to the same task, which is presented in Figure 11. In Figure 12, we showcase some results of combining DreamBooth [34] to include personalized concepts in the generated videos. In Figure 13, we showcase the results of generating videos based on the given first frame by leveraging DDIM inversion [43].

"Two superman are fighting."

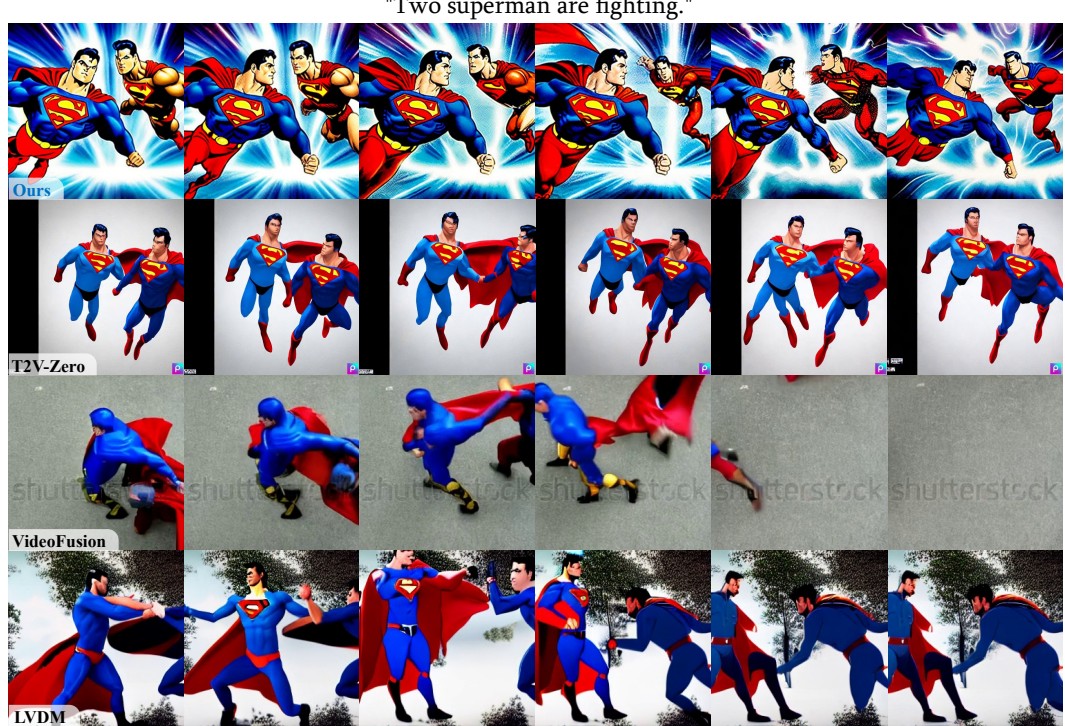

"The growth of a sapling."

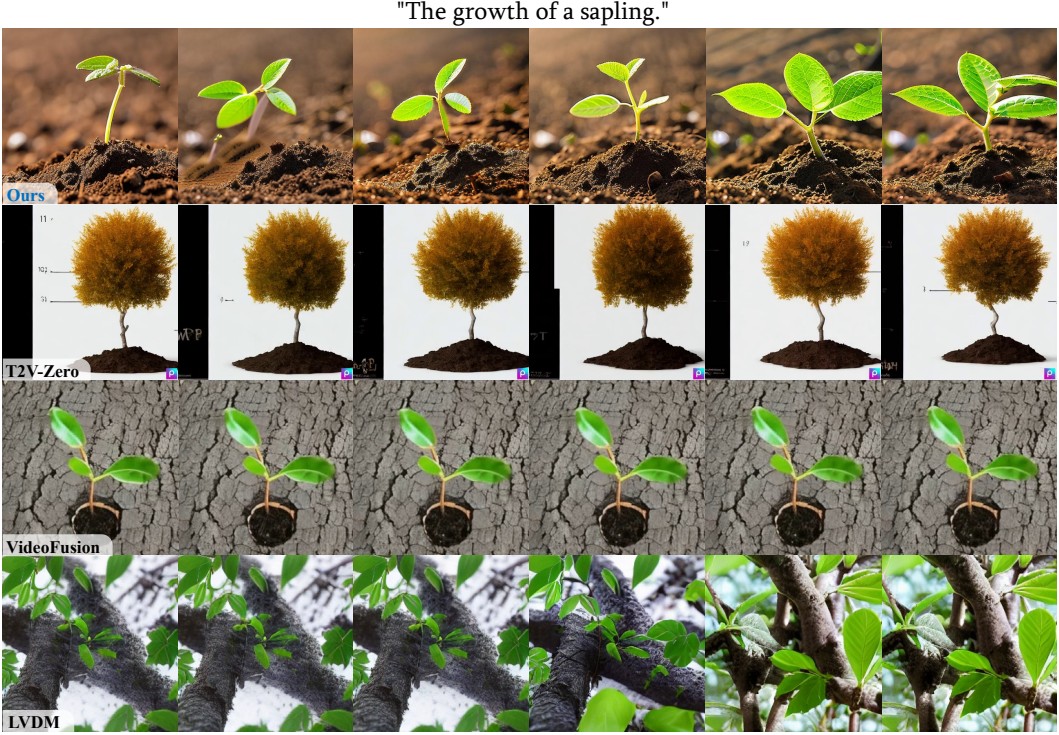

Figure 6: **Additional Qualitative Comparisons.** In the case of *"Two supermen are fighting"*, the LLM vividly decomposes the process of fighting into frames, with the fifth frame depicting *"colliding in a dazzling display of sparks and force"*, which is captured in our result. In the case of *"the growth of a sapling"*, our result clearly presents the gradual sprouting of a small sapling.

"Light a match then the match goes out."

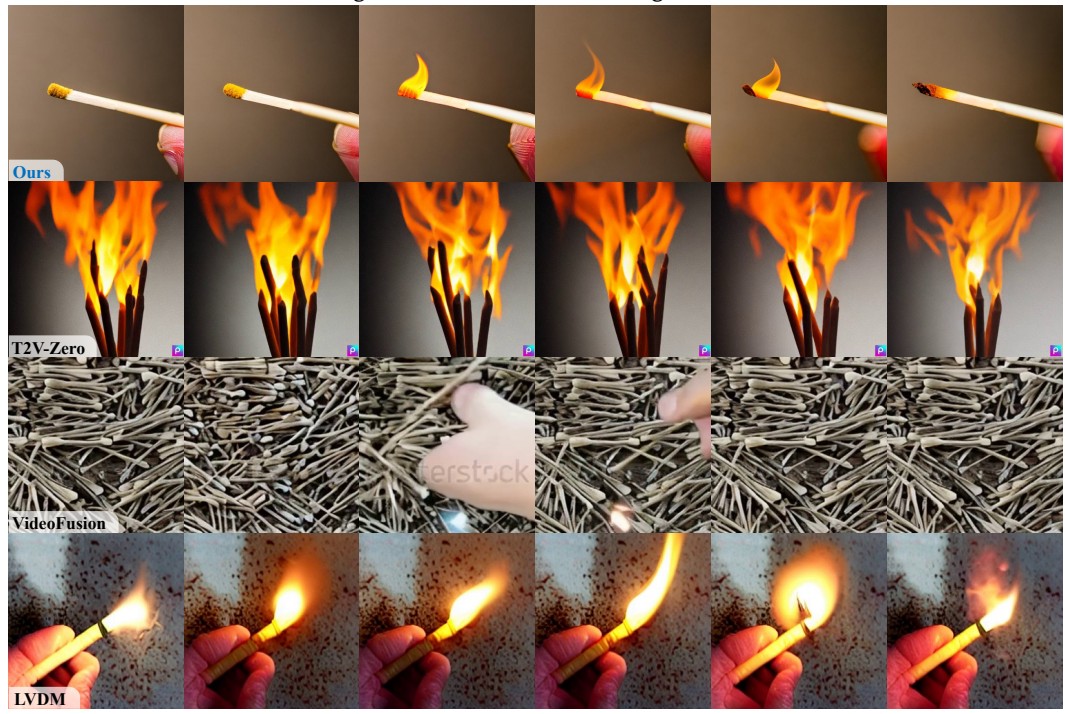

"A teddy bear is greeting."

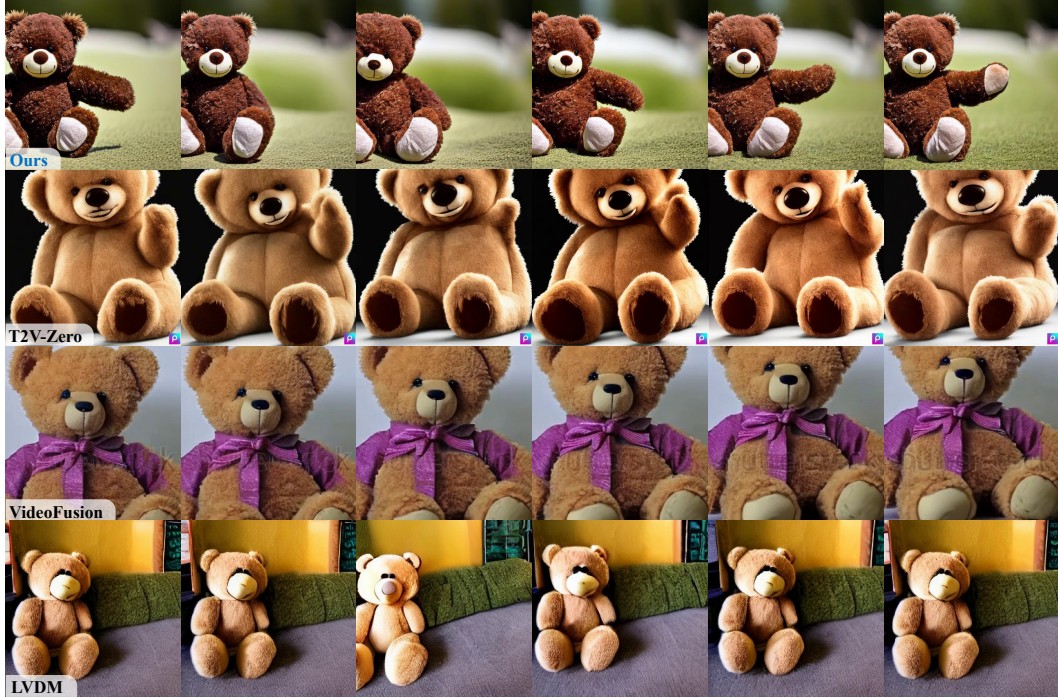

Figure 7: **Additional Qualitative Comparisons.** In the case of *"light a match then the match goes out"*, our method successfully depicts the entire process of a match from lightning, burning to extinguishing. In the case of *"a teddy bear is greeting"*, we exploit the world knowledge of LLM [26] to translate a greeting into a series of specific actions such as waving and smiling.

"A flower gradully blooms."

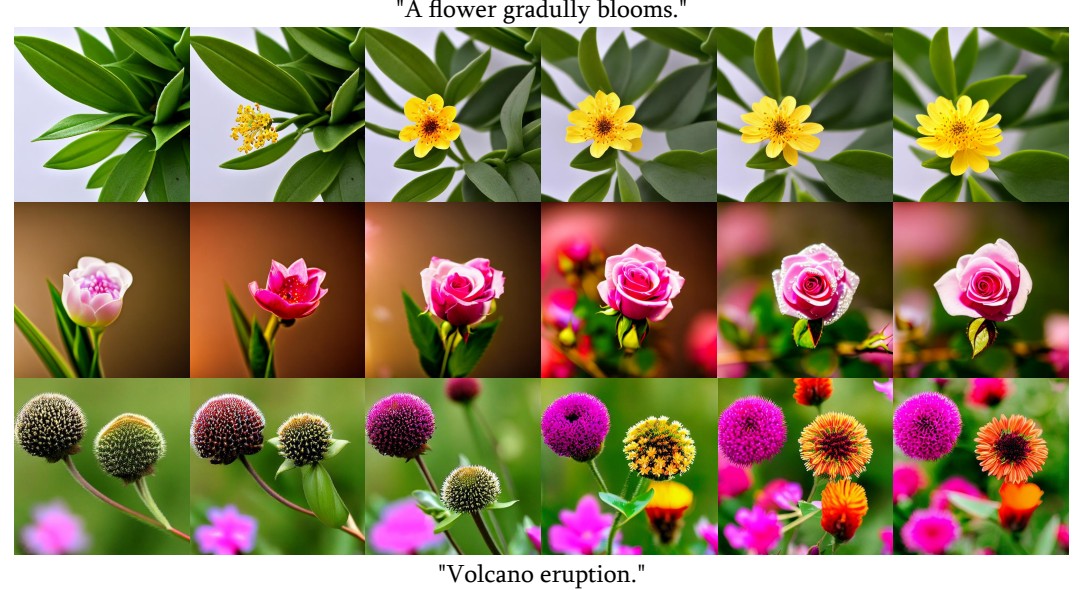

"Volcano eruption."

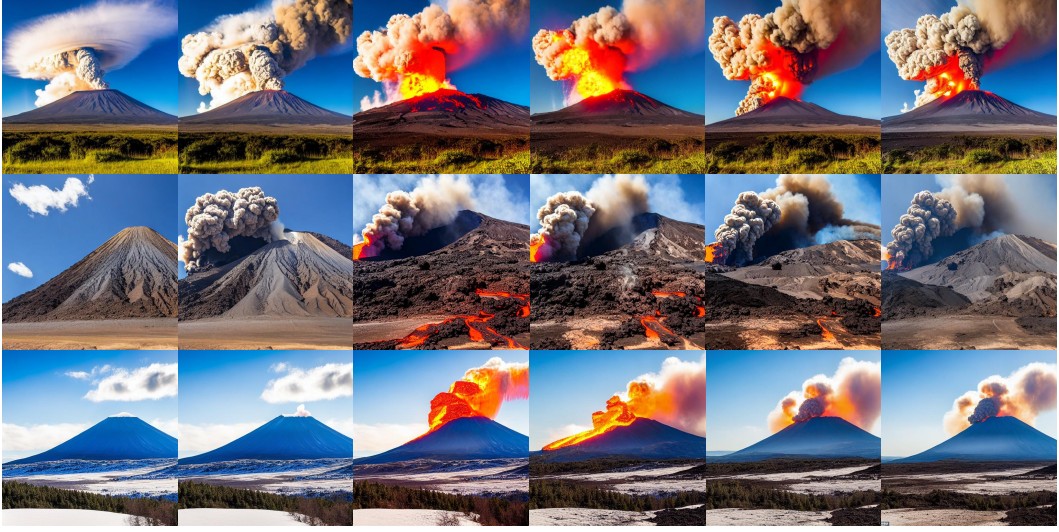

"A dancing mickey."

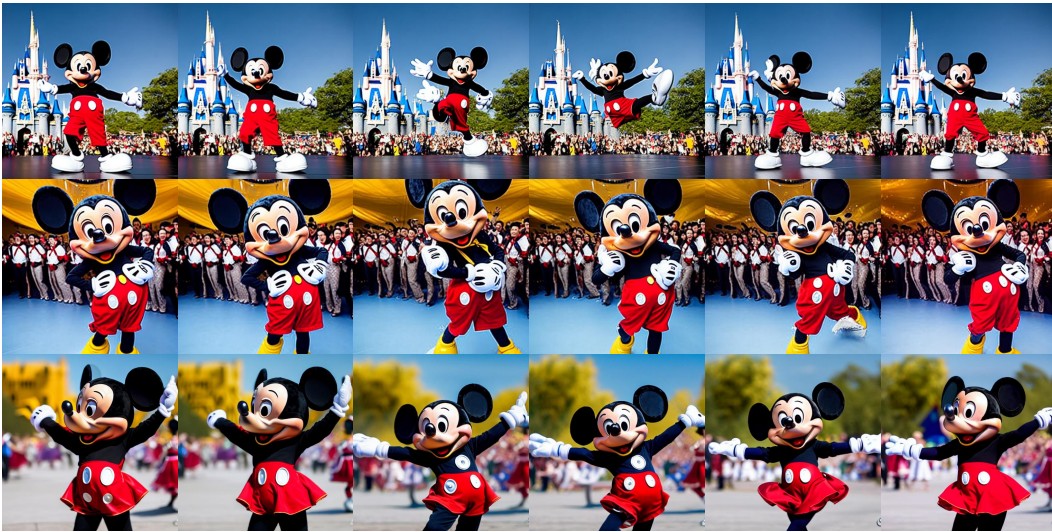

Figure 8: **Additional Qualitative Results.** Multiple results based on the same prompts are shown.

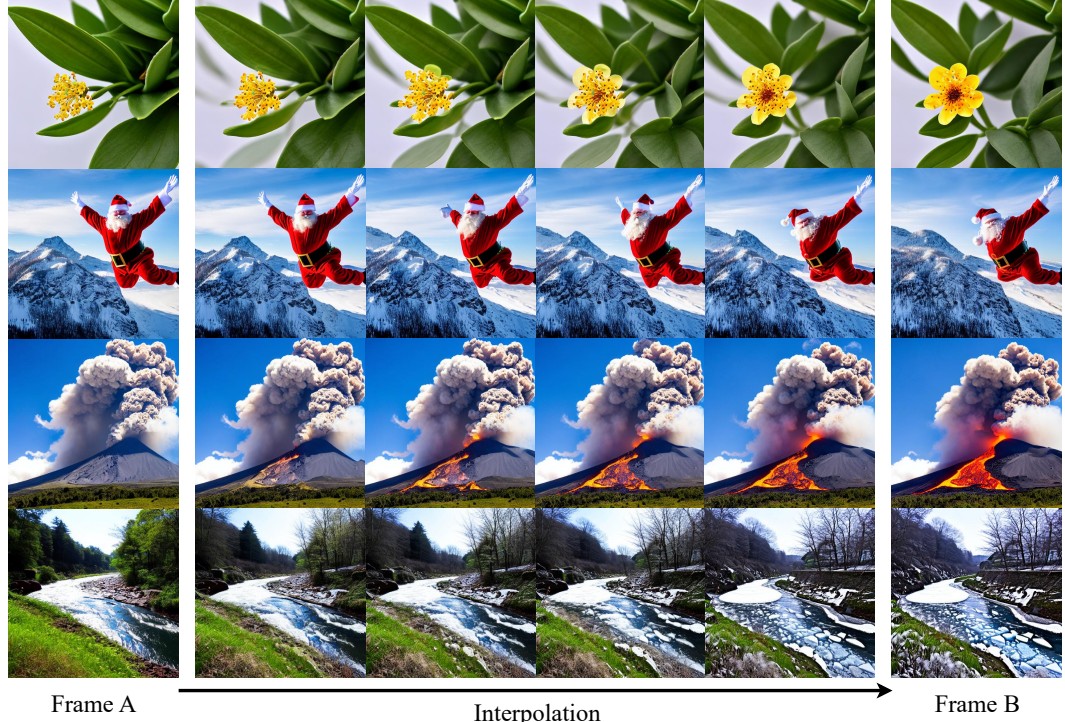

Frame A       Interpolation       Frame B

Figure 9: **Qualitative Results from Interpolation Module.** We interpolate 4 frames between each pair of original neighboring frames. Our interpolation module enables smooth transitions between two key states.

(A) $\cos(\frac{\pi}{2}\lambda)\mathbf{x}_T^{1:f} + \sin(\frac{\pi}{2}\lambda)\boldsymbol{\delta}_T^{1:f}$

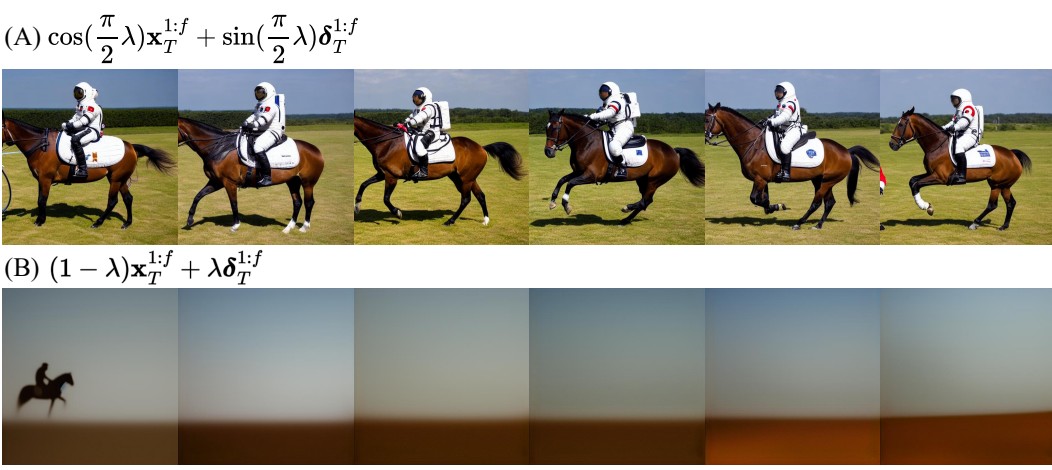

(B) $(1-\lambda)\mathbf{x}_T^{1:f} + \lambda\boldsymbol{\delta}_T^{1:f}$

Figure 10: **Analysis on Joint Noise Sampling.** Without our proposed sampling, the initial noise at each single frame would not follow normal Gaussian distribution, resulting in corrupting frames.

**1st prompt:** "A flower is blooming"

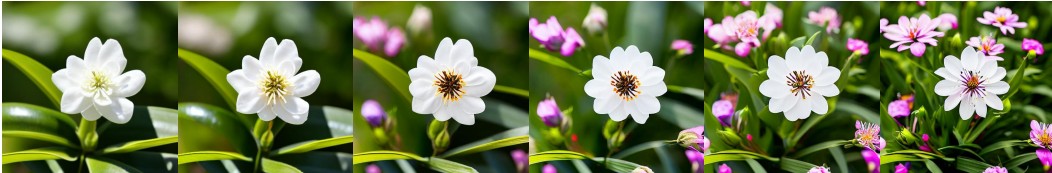

**2nd prompt:** "The flower in the rain"

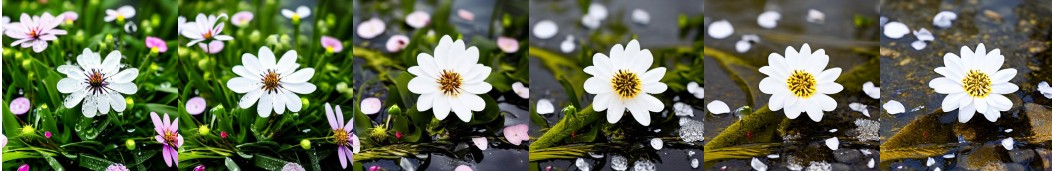

**3rd prompt:** "The flower gradually freezes"

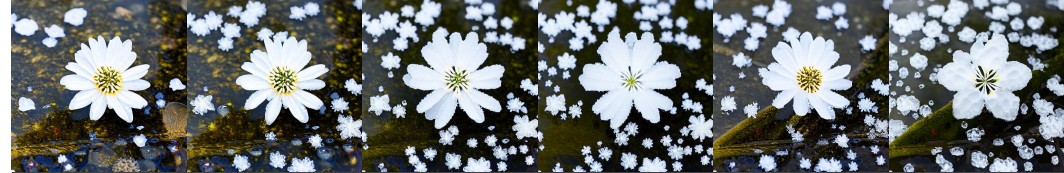

Figure 11: **Extension - Long Video Story.** Our method can generate the long video story based on a sequence of prompts.

"<*ccorgi dog*> is sitting down."

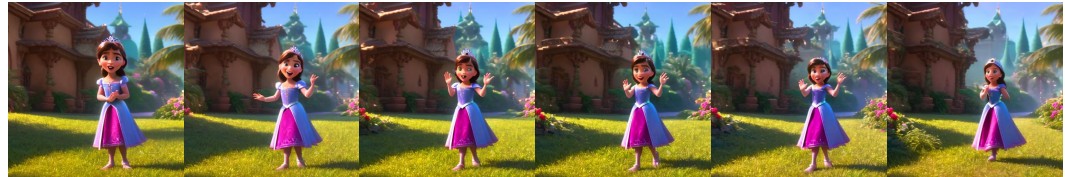

"A princess is waving her hands, <*modern disney style*> ."

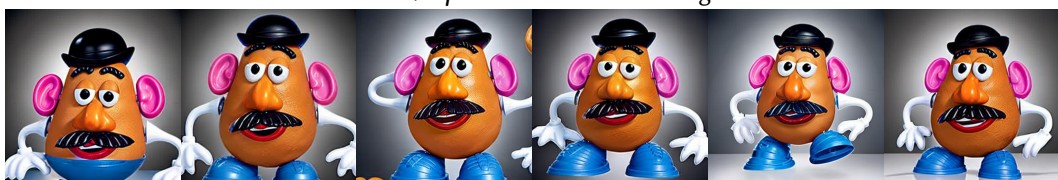

"<*sks mr potato head*> is dancing."

Figure 12: **Extension - Personalization.** Our method can generate videos with user-specific concepts. The tokens of "ccorgi dog", "modern disney style", and "sks mr potato head" are from their respective personalized models ccorgi-dog, Mo di Diffusion, and Mr Potato Head.

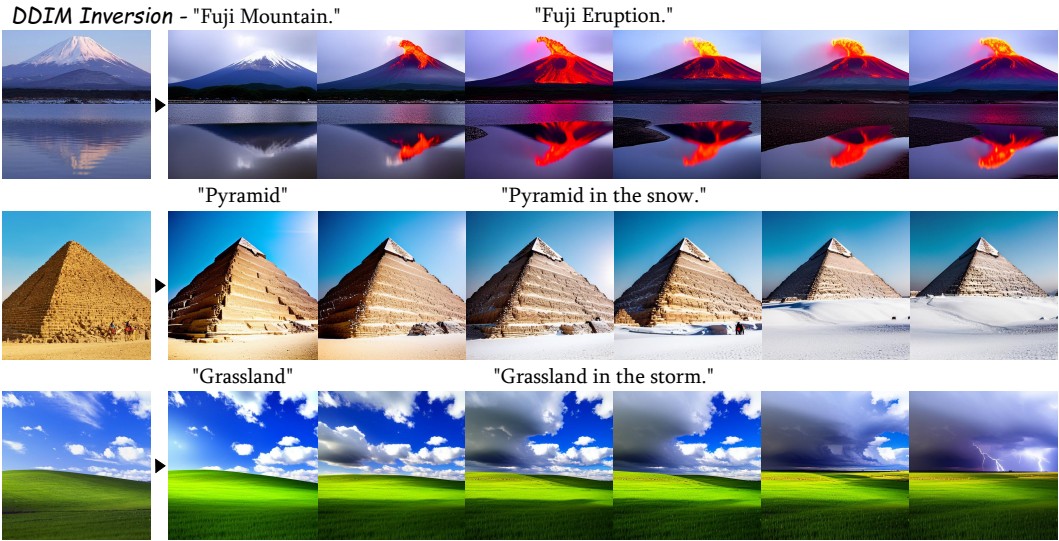

Figure 13: **Extension - Making an Image Move.** Our method can generate videos based on the first frame and its corresponding prompt by combining our method with DDIM inversion [43].

# User Study on Text-to-Video Generation

**Overview:**

Thank you very much for helping us with our research! In this survey, we will present text descriptions along with corresponding videos, which are generated by four state-of-the-art **text-to-video AI models**. Each question will include a text prompt and the corresponding generated videos. Your task is to rank its performance across different dimensions. Additionally, please provide a final comprehensive ranking based on your preferences.

This survey consists of 5 questions and will approximately take 10 min of your time.

Sign in to Google to save your progress. Learn more

\* Indicates required question

Below are four videos generated from "A cluster of flowers blooms"

How would you rate the temporal coherence and smoothness of the videos?    \*
Please assign a score for their **continuity**.

|   | 1 | 2 | 3 | 4 | 5 |
|---|---|---|---|---|---|
| A | ○ | ○ | ○ | ○ | ○ |
| B | ○ | ○ | ○ | ○ | ○ |
| C | ○ | ○ | ○ | ○ | ○ |
| D | ○ | ○ | ○ | ○ | ○ |

Figure 14: Interface of surveys from the user study.

