# OpenReview forum: "Free-Bloom: Zero-Shot Text-to-Video Generator with LLM Director and LDM Animator"
_NeurIPS.cc/2023/Conference — NeurIPS 2023 poster_

### Official Review · Reviewer_6FyB · 2023-07-01

**Soundness:** 3 good
**Presentation:** 3 good
**Contribution:** 3 good
**Rating:** 7
**Confidence:** 4

**Summary:**

This paper presents a novel zero-shot text-video generation method that leverages an LLM as a director to generate per-frame prompts fed into a pre-trained text-image model such as Stable Diffusion. In order to retain temporal coherence, the authors propose (1) a noise distribution consisting of an interpolation of global and local noise, and (2) attend to prior frames in the self-attention layers during the generation process. In addition, they propose a dual interpolation method to generate higher fps video.


**Strengths:**

- The paper is well written and easy to understand
- The presented method is novel and interesting, and outperforms SOTA zero-shot text-video methods, as well as SOTA trained video generation methods in general frame fidelity and visual semantics, while requiring not text-video training itself
- This paper presents an interesting direction in leveraging an LLM’s understanding of the world through language for video generation


**Weaknesses:**

- Quantitative results / evaluation are a little weak, given the relatively small sample size (20 prompts, 4 videos each)
- Results of ablations are shown qualitatively, and could be improved more through more rigorous quantitative evaluations


**Questions:**

- How is $\lambda$ chosen during generation? Is it fixed, or does it change over the video timestep?
- How does this method behave in scenes with more content? (e.g. more people, more animals). Can the LLM directory + SD model distinguish specifications of motions between two humans doing different things?
- How about scenes with larger shifts in motion (e.g. a camera panning to the left)? As in this case frame 0 (concatenated to the self-attention) would be less useful.
- Or more complex motion that may be difficult to describe in words (e.g. each step in a dance)?


**Limitations:**

Yes, the authors discuss limitations

---

> ### Author Rebuttal · Authors · 2023-08-10
>
> > **Benchmark**:  Quantitative results/evaluation are a little weak, given the relatively small sample size.
>
> Thanks for pointing out this. The standard benchmarks for video generation, especially for zero-shot video generation, are still building up in the community. The videos in our benchmark are diverse, including Text2Video-Zero prompts and prompts with rich semantic content (such as how to showcase flower blooming, saying hello, ice melting, etc.).
>
> Here, we enlarge our benchmark to 68 prompts with 4 videos each due to time limitations. Note that both the quantity and diversity of videos in our benchmark exceed that in previous methods.
>
> | Method | Training-Free | CLIP Metrics↑| Fidelity↑ | Temporal↑ | Semantic↑ | Rank↓ |
> | --- | :---: | :---: | :---: | :---: | :---: | :---: |
> |VideoFusion [1]| - | 0.471| 3.386 | 3.914 | 3.209 | 2.368 |
> | LVDM [2] | - | 0.476 | 3.276 | 3.674 | 3.206 | 2.558 |
> | T2V-Zero [3] | √ | 0.482 | 3.469 | 2.721 | 3.003 | 3.033 |
> | ours | √ | 0.471/0.483* | 4.185 | 3.318 | 3.909 | 2.042 |
>
> [1] Luo, Zhengxiong, et al. "VideoFusion: Decomposed Diffusion Models for High-Quality Video Generation." *Proceedings of the IEEE/CVF Conference on Computer Vision and Pattern Recognition*. 2023.
>
> [2] He, Yingqing, et al. "Latent video diffusion models for high-fidelity video generation with arbitrary lengths." *arXiv preprint arXiv:2211.13221* (2022).
>
> [3] Khachatryan, Levon, et al. "Text2video-zero: Text-to-image diffusion models are zero-shot video generators." *arXiv preprint arXiv:2303.13439* (2023).
>
> > **Quantitative ablations**:  Results of ablations are shown qualitatively, and could be improved more through more rigorous quantitative evaluations
>
> Thanks for your valuable feedback. Per your advice, we conducted additional ablation studies and evaluated their performance in both human evaluation and automatic metrics. As presented in all dimensions, the absence of either the noise sampling module or the attention shifting module would significantly degrade the performance, and the lack of both would result in extremely poor results.
> |Serial Prompting|Joint Noise Smapling|Shifting Self-attention|CLIP Metrices↑|Temporal↑|Identical↑|Rank↓|
> |:---:|:---:|:---:|:---:|:---:|:---:|:---:|
> |√|-|-|0.4429 / 0.4661|1.62|1.95|3.66|
> |√|√|-|0.4482 / 0.4712|1.83|2.30|3.04|
> |√|-|√| 0.4509 / 0.4736|2.52|2.71|2.19|
> |√|√|√| 0.4521 / 0.4773|3.67|3.46|1.11|
> > $λ$: How is $λ$ chosen during generation? Is it fixed, or does it change over the video timestep?
>
> The hyperparameter setting of $λ=0.2$ is recommended for most cases. $λ$ is fixed and does not change over the video timestep. $\lambda$ is used in the joint noise sampling module to regulate the ratio between the independent noise and the united noise. A video’s per-frame noise weighs from the frame-wise independent noise and one unified video noise, where the weights of the two noises are the same across different frames.
> >**Scenes with more content**: How does this method behave in scenes with more content? (e.g. more people, more animals). Can the LLM directory + SD model distinguish specifications of motions between two humans doing different things?
>
> We supplemented some additional examples of scenes with more subjects in Figure 5 in the one-page PDF. As shown in the case “a dog is running and then another dog joins”, our pipeline performs well on simple scenarios involving two or three objects performing similar actions.
>
> However, when involving complex scenes with different subjects with diverse features performing distinct actions, such as shown in the case “A man is dancing and a woman is watching”, our method does indeed face limitations. In our practice, LLMs can separate and describe the detailed actions of subjects within each sentence in a reliable manner. In contrast, the current version of LDMs struggles to associate the action with the corresponding subject. If a more powerful LDM emerges in the future, we believe our pipeline holds the potential to achieve this as well.
> >**Scenes with larger shifts in motion**: How about scenes with larger shifts in motion (e.g. a camera panning to the left)? As in this case frame 0 (concatenated to the self-attention) would be less useful.
>
> Thank you for considering more general applications and scenes. Unfortunately, this challenging camera panning lies beyond the scope and capability of our research, posing a significant challenge even for models trained on massive video data.
>
> The reason is that when involving camera panning, the model is needed and expected to construct from or at least have the ability to image a 3D scene from one single frame, which is very challenging. Also, the current LDMs remain unresponsive to layout descriptions, hindering the realization of scene shifting even with detailed prompts like “subject on the left”, ”… in the middle”, ”… on the right”, etc.
>
> >**Complex motion in words**:  Or more complex motion that may be difficult to describe in words (e.g. each step in a dance)?
>
> Our pipeline is capable of generating corresponding results for some straightforward action sequences, such as raising and lowering the right hand, lifting the left foot, and similar motions. Figure 6 in the one-page PDF shows these cases.
>
> But, our method indeed is limited in cases of intricate dance movements, where the pose details are required. On the one hand, LLMs often fail to provide prompts detailed at fine granularity enough, which is also the limitation of using natural language as the condition. For example, for the input “a man is hip hop dancing”, the LLMs are likely to provide vague and high-level state descriptions like “does the ‘wave’ by creating a ripple effect through his body”. On the other hand, the LDMs cannot generate images that match prompts with highly detailed descriptions. How to make LDMs comprehend complex prompts, including spatial directions, colors, and textual information, remains a prominent research focus in the T2I diffusion area.

---

> > ### Comment · Reviewer_6FyB · 2023-08-10
> > **Response**
> >
> > Thank you for your clarifications and further evaluations. My main concern regarding evaluation was addressed, so I will raise my score to a 7.

---

> > > ### Author Response · Authors · 2023-08-11
> > > **Reply to Reviewer 6FyB: Thank you!**
> > >
> > > We thank the reviewer for the quick reply and consideration! It is heartening to receive this positive evaluation. Your detailed review and feedback are valuable to us in further shaping our work. Once again, we express our gratitude for your time and appreciation!

---

### Official Review · Reviewer_yWMb · 2023-07-05

**Soundness:** 3 good
**Presentation:** 3 good
**Contribution:** 2 fair
**Rating:** 5
**Confidence:** 5

**Summary:**

This paper proposes a novel approach to achieve Text-to-Video generation through Zero-Shot learning using LLMs and a diffusion-based image generation model. By using LLMs to generate detailed and varied descriptions for each frame of the video, the proposed method captures the changes in the video frames. The generated video frames are then constrained by self-attention module in SD and input noise to align with the input text and maintain a certain level of coherence. This paper also proposes a zero-shot frame interpolation algorithm based on a diffusion-based image generation model. While the current results may not be very satisfactory, this approach is a low-cost and Zero-Shot learning method that does not require additional training and can directly utilize existing SD models.

**Strengths:**

1.The entire process is Zero-Shot learning, without the need for any additional training, making the method low-cost. Moreover, compared to existing zero-shot video generation algorithms, the proposed approach has achieved significant improvements in performance.
2.This algorithm effectively combines two pre-trained models that typically require significant computational resources for training and easy to follow.
3.Since there is no fine-tuning or post-pretraining of the existing powerful image generation model, the proposed method preserves the ability of the model to generate high-quality images to the maximum extent. Compared to models trained on video data, the image quality of the generated frames is better.

**Weaknesses:**

1.Lacking a comparison of the quality of video generation guided by different large language models. Only ChatGPT was used in the experiments.
2.The computational and time overhead during inference may increase when using LLMs.
3.The proposed frame interpolation algorithm for video frames was not compared numerically with other state-of-the-art algorithms on the test set in the article.
4.The ablation study only provides qualitative analysis and does not include quantitative comparisons.

**Questions:**

1.What is the test set in Table 1? I did not find any relevant explanation in the paper.
2.I didn't understand Figure 5, and what is the variable 𝜏* referring to?

**Limitations:**

1.The proposed pipline was not trained on video data, so compared to methods trained on videos, it is still weaker in terms of temporal consistency of the generated video frames.
2.The proposed method is largely limited by the high-order reasoning capability of LLMs for video imagination and the image generation capability of the diffusion-based image generation model.

---

> ### Author Rebuttal · Authors · 2023-08-10
>
> > **Comparion with different LLMs**: Lacking a comparison of the quality of video generation guided by different large language models. Only ChatGPT was used in the experiments.
>
> Thanks for your valuable feedback. Per your advice, we conduct experiments using other LLMs (GPT-4 and Bard), and the results validate that our methods are very robust to these LLMs. Due to limited space, we only show the case "a flower is gradually blooming" in Figure 4 in the one-page PDF. And six per-frame prompts of every LLM are given below. In practice, LLMs output prompt sequences to provide semantic guidance for frames. As long as the semantic guidance is reasonable and coherent, our proposed modules on the LDM can ensure the generation of sensible video.
>
> ```
> GPT-3.5:
> > A close-up view of a green stem with buds tightly closed, waiting to bloom.
> > The buds slowly unfurl, revealing delicate petals in vibrant colors.
> > The flowers continue to bloom, forming a beautiful cluster of various shapes and hues.
> > A gentle breeze sways the blossoms, creating a mesmerizing dance of colors.
> > The cluster of flowers is now fully open, displaying a stunning arrangement of shapes and textures.
> > As the sunlight bathes the blooms, they glisten with dewdrops, exuding a sense of freshness and vitality.
> GPT-4:
> > A tight bud is suspended in the center of the frame against a background of soft pastels.
> > Small, delicate cracks appear on the surface of the bud, revealing a hint of white color within.
> > The bud begins to unfurl, petals delicately stretching outward, their hue a pure white.
> > The bloom is halfway open, the curling petals forming an intricate pattern, sunlight shimmering off their velvety surface.
> > The flower is almost fully bloomed, the vibrant petals dominating the frame, contrasting starkly with the greenery in the background.
> > The flower is in full bloom, petals splayed out in a radiant display, revealing a golden heart of stamens in the very center.
> Bard:
> > A close-up of a white flower bud, tightly closed.
> > The flower bud begins to open, revealing a few white petals.
> > The flower continues to open, more petals unfolding.
> > The flower is now fully open, with a delicate white bloom.
> > The flower begins to wilt, the petals starting to curl.
> > The flower is now completely wilted, the petals falling to the ground.
> ```
>
> > **The computational and time overhead** during inference may increase when using LLMs.
>
> Thanks for pointing out this limitation. We agree that the cost of LLMs should be taken into consideration. However, LLMs' inference is fast currently (considering it can engage in real-time conversations with ChatGPT), which is a small overhead compared to the entire pipeline since the LDM is relatively slower for generating even a single image (20-30s).
>
> > **Quantitative comparison of interpolation**: The proposed frame interpolation algorithm for video frames was not compared numerically with other state-of-the-art algorithms on the test set in the article.
>
> Per your feedback, we conduct experiments on our frame interpolation algorithm on both human evaluation and automatic metrics. For the Image CLIP score, we average the CLIP similarity of the image features of every two adjacent frames. In the user study, we ask raters to compare the two videos from different methods. Our method surpasses another SOTA approach in all dimensions, particularly in terms of individual frame quality.
>
> ||Image CLIP Score|Temporal|Fidelity|Rank|
> |:---:|:---:|:---:|:---:|:---:|
> |Ours Vs AMT [1]|74.55%|74.07%|81.48%|74.07%|
>
> [1] Li, Zhen, et al. "AMT: All-Pairs Multi-Field Transforms for Efficient Frame Interpolation." *Proceedings of the IEEE/CVF Conference on Computer Vision and Pattern Recognition*. 2023.
>
> > **Quantitative ablation study**: The ablation study only provides qualitative analysis and does not include quantitative comparisons.
>
> Thanks for your valuable feedback. Per your advice, we conducted additional ablation studies and evaluated their performance in both human evaluation and automatic metrics. As presented in all dimensions, the absence of either the noise sampling module or the attention shifting module would significantly degrade the performance, and the lack of both would result in extremely poor results.
> |Serial Prompting|Joint Noise Smapling|Shifting Self-attention|CLIP Metrices↑|Temporal↑|Identical↑|Rank↓|
> |:---:|:---:|:---:|:---:|:---:|:---:|:---:|
> |√|-|-|0.4429 / 0.4661|1.62|1.95|3.66|
> |√|√|-|0.4482 / 0.4712|1.83|2.30|3.04|
> |√|-|√| 0.4509 / 0.4736|2.52|2.71|2.19|
> |√|√|√| 0.4521 / 0.4773|3.67|3.46|1.11|
> > **Test set**: What is the test set in Table 1? I did not find any relevant explanation in the paper.
>
> Thanks for pointing out this. We use prompts from the webpage of Text2Video-Zero, and we add more prompts that incorporate more complex event content. Our motivation is to validate whether our method achieves better results in the existing cases but also has an impressive effect on cases with rich semantic content (such as how to showcase flower blooming, saying hello, ice melting, etc.). The quantity and diversity of videos in our test set exceed that of the current methods for zero-shot video generation.
>
> > $τ^*$: I didn't understand Figure 5, and what is the variable 𝜏* referring to?
>
> $τ^*$ is the hyperparameter in the interpolation module, which is the step threshold of changing the weight from focusing on contextual frame paths to the self-denoising path. Figure 5 illustrates the influence of the selection of $τ^*$. We sincerely apologize for the oversight in Figure 5, where the image for $τ^*$ approaching 1000 and $τ^*$approaching 1 are switched.
>
> We defined our interpolation step in Eq.8, in which $m(t)$ varies with denoising timesteps and controls the weight of two paths. To make the approach simpler, we set $m(t)$ as a step function empirically in the experiments. It can be formulated as $m(t)=0.1, t\geqτ^*; m(t)=1,t<τ^*$ (larger t indicates earlier denoising stage).

---

> > ### Comment · Area_Chair_nqMo · 2023-08-18
> >
> > Dear Authors,
> >
> > Thank you for your message and we will take your response into consideration.
> >
> > Best,

---

### Official Review · Reviewer_mgrm · 2023-07-06

**Soundness:** 4 excellent
**Presentation:** 4 excellent
**Contribution:** 3 good
**Rating:** 7
**Confidence:** 4

**Summary:**

In this work, the authors propose a principled pipeline for text to video generation. Specifically, three techniques are proposed: (1) a LLM based per-frame prompt generation, so that the motion/dynamics of each frame can be better specified. (2) a noise joint sampling schedule, and a step-aware attention shift is proposed to enhance temporal consistency. (3) an interpolation module to generate longer videos. Extensive experiments and results have demonstrated the effectiveness of the proposed work.

**Strengths:**

1. The pipeline is technically sound and generally makes sense to me. It is straightforward to use the strong prior from pretrained LLM to give more temporal/motion/appearance information to each frame, to facilitate video generation. The design of noise scheme, sampling, and attention shift is technically sound.

2. The exposition of this paper is very good and clear to me.

**Weaknesses:**

1. In my honest opinion, directing comparing this work w/ many baselines is kind of unfair, since this involves strong prior in LLM, and this is the key factor of improvement in video generation. I strongly recommend the authors to compare w/ some similar text2video pipelines that also factorize the unified single prompt condition into per-frame motion-aware conditions. For example, first generate a sequence of optical flow maps, then takes them as conditions to generate the video.

2. Since I'm not the expert in NLP domain, I'm concerned about the ability of LLM to generate very reasonable or temporally-coherent per-frame prompts. I'm also curious about the feature trajectory of such text features. Say, you generate the per-frame prompt of p1, p2, p3, ..., pn, and use the text encoder to get text features f1, f2, ..., fn for cross-attention. Do they form a smooth trajectory in the high-dimensional text feature space? Do the authors think that, a smooth feature transition among frames can guarantee a smooth video? If so, a naive idea for improvement of this work is to add some regularizations to constrain the smoothness of text features across frames.

**Questions:**

elaborated in the above sections.

---

> ### Author Rebuttal · Authors · 2023-08-10
>
> > **Prior in LLM (part 1)**: To my honest opinion, directing comparing this work w/ many baselines is kind of unfair, since this involves strong prior in LLM, and this is the key factor of improvement in video generation.
>
> As the reviewer pointed out, using LLMs introduces prior knowledge about how the visual content evolves throughout the frames according to the unified prompt. One of our contributions is to propose this pipeline to leverage this prior in the text-to-video generation.
>
> However, language is sometimes not as exact as other conditions, such as pose. For example, it is hard to use language to precisely describe where a person's pose skeleton and the positions in a picture. From this aspect, we consider that the knowledge from LLMs could be a kind of weak prior.
>
> Also, although LLM brings a prior, it also poses a hard challenge: given per-frame fine-grained prompts, how to adapt LDMs to produce a coherent and contextually fitting frame sequence. Per-frame prompts lead to significant incoherence in the LDM’s frame results, and language alone can not guide LDMs on how to make each adjacent frame coherent.
>
> > **Prior in LLM (part 2)**: I strongly recommend the authors to compare w/ some similar text2video pipelines that also factorize the unified single prompt condition into per-frame motion-aware conditions. For example, first generate a sequence of optical flow maps, then takes them as conditions to generate the video.
>
> Thank you for this advice. We reconducted a survey of recent works, but unfortunately, we did not find a zero-shot work capable of giving a text input and generating per-frame motion flow or per-frame optical flow. Actually, our pipeline achieves text-to-per-frame condition generation and then video generation. We humbly ask for your kind advice if we have missed some work.
>
> As an alternative, we compare our method against Tune-a-Video, which introduces a reference video as a much stronger prior than ours. Figure 2 in the one-page PDF shows that Tune-a-Video benefits from the strong prior but is also highly limited by the strong bias from the reference video, failing to generate videos of specified subjects yet similar and consistent movements. On the contrary, our method can generate diverse videos corresponding to the input text.
>
> > **The ability of LLM (part 1)**: Since I'm not the expert in NLP domain, I'm concerned about the ability of LLM to generate very reasonable or temporally-coherent per-frame prompts.
>
> In our practice, the LLMs can actually produce reasonable and temporally-coherent per-frame prompts because LLMs can learn common behaviors, phenomena, and events as well as their co-occurrence relationships, sequence patterns, and semantic associations from large-scale text data. All the latest LLMs we experiment with, such as GPT-3.5, GPT-4, bard, etc., are capable of decomposing actions like ”jumping into the water“ and translating "say hello" into waving hands and smelling. Our user study also suggests that raters perceive almost all video content trajectories as reasonable.
> Certainly, LLMs do have their limitations: they cannot provide guidance as precise as other strong conditions, such as pose sequences, as mentioned above. However, we claim this is also a benefit to keep the flexibility and generalization to generate a variety of videos from one given prompt. Coherence of details not provided by LLMs can be constrained and improved by our other three techniques, i.e., joint noise sampling, attention shift, and dual-path interpolation.
>
> > **The ability of LLM (part 2)**: I'm also curious about the feature trajectory of such text features. Do the authors think that, a smooth feature transition among frames can guarantee a smooth video? If so, a naive idea for improvement of this work is to add some regularizations to constrain the smoothness of text features across frames.
>
> Thank the reviewer for this suggestion! We consider that all text features in one video only need to be relatively close (or similar) to each other. In our practice, most of the per-frame text features generated by ChatGPT remain relatively close to each other. Figure 3 in the one-page PDF shows average CLIP similarities (average on all cases in the test set) between six text features in each video are high. We have also applied this characteristic in our interpolation module, where we perform linear interpolation on the adjacent text features to acquire in-distribution text features for the interpolated frame.
>
> Thanks to the high similarities and our proposed techniques on sampling and attention to encourage LDM to output smooth frame sequences, text features are not required to form a smooth trajectory.

---

> > ### Comment · Reviewer_mgrm · 2023-08-10
> > **Thanks for the detailed response**
> >
> > I thank the authors for the detailed response, which has addressed most of my concerns!
> >
> > In terms of text-flow-video papers, maybe they are some concurrent NeurIPS submissions, so it's unfair to ask the authors to compare w/ them. I apologize for asking about this in the initial review.
> >
> > Actually, all the NeurIPS submissions in my batch of 5 papers are about DM-based text-to-video generation, and imho, this paper's technical soundness, exposition, and contribution could be the best. So I'm raising my score to 7 and vote for the acceptance of this paper.

---

> > > ### Author Response · Authors · 2023-08-11
> > > **Reply to Reviewer mgrm: Thank you!**
> > >
> > > Thank the reviewer for the appreciation and engagement in assessing our work! We are grateful for the reviewer’s understanding regarding the challenge in comparison. We highly agree with the consideration of this aspect that once the concurrent works are made available, we will study, compare, and analyse accordingly. Once again, we express our gratitude for the time and prompt reply!

---

### Official Review · Reviewer_VS1k · 2023-07-06

**Soundness:** 2 fair
**Presentation:** 2 fair
**Contribution:** 2 fair
**Rating:** 5
**Confidence:** 4

**Summary:**

The paper proposes a zero-shot text-to-video generating pipeline called Free-Bloom, which first use LLM to generate prompt sequence decribing frames in a video, then generate frames according the prompts. To enhancing coherence, the authors proposed joint noise sampling, step-aware attention shift and dual-path interpolation. The authors compare their method with other video generator quantitatively (Clip metrics, user study) and qualitatively.

**Strengths:**

1. explore zero-shot text-to-video generation LLM director, which leverage the story generation ability of LLM to generate semantic meaningful frame sequences.
2. Good per-frame quality.
3. tried several methods to enhancing coherence of zero-shot video generation.

**Weaknesses:**

1. insufficient ablation study.

   a. there's only two qualitative cases in ablation study, which is not convincing enough. More cases and quantitative results (e.g. CLIP metrics) should be provided.

   b. given that joint noise sampling, step-aware attention shift and dual-path interpolation are universal technics for zero-shot video generation, experiment result of combining these technics and past zero-shot methods (e.g. text2video-zero) is needed.

2. The coherence is not good enough. e.g. sudden changes in the color / indentity can be observed. There is a fatal flaw in technical design: authors set m(t) to 1 and discard attention shift when t is small, which may lead to strong incoherence.

3. Insufficient clarification of experiments, e.g. prompts to LLM, hyperparameters.

**Questions:**

1. Please provide more clarification of experiments, including prompt engineering to get frame description, hyperparamters (such as \tau, \tau*)

2. The coherence is not good, but the idea of leveraging LLM is interesting. Is it possible to combine LLM director with other existing pretrained text-to-video methods (e.g.  make-a-video) to enhance conherence?

**Limitations:**

see weakness and questions.

---

> ### Author Rebuttal · Authors · 2023-08-10
>
> > **Insufficient ablation study (part 1)**: more cases and quantitative results should be provided.
>
> Thanks for your valuable feedback. Per your advice, we conducted additional ablation studies and evaluated their performance in both human evaluation and automatic metrics. As presented in CLIP metrics and all dimensions in the user study, the absence of either the noise sampling module or the attention shifting module would significantly degrade the performance, and the lack of both would result in extremely poor results.
> |Serial Prompting|Joint Noise Smapling|Shifting Self-attention|CLIP Metrices↑|Temporal↑|Identical↑|Rank↓|
> |:---:|:---:|:---:|:---:|:---:|:---:|:---:|
> |√|-|-|0.4429 / 0.4661|1.62|1.95|3.66|
> |√|√|-|0.4482 / 0.4712|1.83|2.30|3.04|
> |√|-|√|0.4509 / 0.4736|2.52|2.71|2.19|
> |√|√|√|0.4521 / 0.4773|3.67|3.46|1.11|
> > **Insufficient ablation study (part 2)**: experiment result of combining these proposed technics and past zero-shot methods (e.g. text2video-zero) is needed.
>
> We combine our technics with text2video-zero (T2VZ) and show the results in Figure 1 of the one-page PDF.
>
> - **Joint Sampling w/ T2VZ:** we substitute the initial noise of T2VZ with our sampling method. Compared with the T2VZ’s original noise that adds motion dynamics to the base latent code, our sampling method can ensure the quality of generated individual frames remains dependable even as the number of frames increases.
> - **Attention Shift w/ T2VZ:** we re-implemented the attention control in T2VZ. The figure shows that under the scheme proposed by T2VZ, our step-aware attention shift can achieve slightly better results than T2VZ's attention strategy. But, we still want to claim that our attention shift's contribution cannot be fully reflected in this experiment, that is, working much better for challenging cases with more complex motions and frame-wise prompts than "moving image" in T2VZ. When all the frames are generated based on one single prompt, it is more like creating "a trembling gif", leading attention to previous frames and current frames less critical.
> - **Interpolation w/ T2VZ:** we directly apply our interpolation algorithm to the generated video of T2VZ. Our interpolation module can supplement intermediate states between two frames in the original video, thereby increasing the frame rate.
>
> > **The coherence is not good enough (part 1).**
>
> We have significantly improved the temporal coherence compared to previous zero-shot T2V methods, and our semantic coherence is much better than both zero-shot and trained T2V methods, according to our user study.
>
> We appreciate the reviewer for pointing out the coherence concern, and the coherence in video generation is essential but incredibly challenging in a zero-shot setting without any reference videos. Many of the efforts in our work revolve around coherence, and our new technicals indeed improve coherence a lot and can be a good starting point for future research to enhance coherence further.
> > **Coherence (part 2)**: Sudden changes in the color/indentity can be observed. There is a fatal flaw in the technical design: authors set m(t) to 1 and discard attention shift when t is small, which may lead to strong incoherence.
>
> This is not a design flaw. The $m(t)$ influences interpolated frames but is irrelevant to the contextual frames in the low-frame-rate stage. Thus, it is unrelated to the identical/coloring coherence.
>
> Also, attention shift is not discarded in the interpolation process (equation 8), whether t is small or large. The term $\tilde{\mathbf x}_t^f$ after $(1-m(t))$ is derived from the latents of contextual frames, and the latents are obtained through their attention-shifting on their paths. And the term following $m(t)$ pertains to the frame denoising path, on which the attention-shift operation is also performed. Notice that smaller $t$ indicates the later denoising stage, thereby needing to focus more on the interpolated frame's denoising path. So, we set $m(t)$ as a step function empirically to simplify the pipeline and attain straightforward results.
>
> Certainly, if we define $m(t)$ as a more elegant function varying through time steps, the performance of interpolation may be further enhanced.
> > **Insufficient clarification of experiments**, e.g. prompts to LLM, hyperparameters (such as \tau, \tau*).
>
> **Prompts to LLMs:** Please refer to Appendix Section C.1.
>
> The settings of the **hyperparameters** $τ=400$  and $τ^*=400$ (when timestep T=1000) are recommended for most practical use cases.  According to specific cases, they could be additionally adjusted.
>
> - For cases involving more semantic and motion changes, as exemplified by “Teddy bear jumps into water” in Figure 3, it is advisable to opt for larger values of $τ$ and $τ^*$. This choice put more emphasis on the current frame by shifting attention to the current frame at an earlier step ($τ$) and increasing the weight of the self-denoising path at an earlier step ($τ^*$).
> - For cases with basically similar frames, such as “Light a match then the match goes out” in Appendix Figure 2, smaller $τ$ and $τ^*$would be suitable.
>
> > Is it possible to **combine LLM director with other existing pretrained text-to-video methods** (e.g. make-a-video) to enhance coherence?
>
> We agree that leveraging pre-trained text-to-video methods can improve the coherence of generated videos, as training with an immense amount of video samples allows the models to acquire knowledge about how to maintain coherence between frames.
>
> However, it is not easy to combine LLM director with pre-trained T2V methods in a training-free manner because videos generated by pre-trained T2V models are likely to conflict with the per-frame prompts. Addressing the conflict is a promising direction but is currently out of this work's scope. On the other hand, if requiring additional training, it is against the initial motivation of our present work considering the data- and cost-efficient.

---

> > ### Comment · Reviewer_VS1k · 2023-08-17
> >
> > Thanks for your reply, especially the additional ablation study!
> > I am still a little bit confused about Coherence (part 2):
> > 1. Is attention-shifting only applied in Video Generation Stage, but not applied in Interpolation Empowerment Stage, according to the caption of Figure 2?
> > 2. Assume that attention-shifting is also used in Interpolation Empowerment Stage. Considering (a) attention-shifting is not used when t < τ  (Equation 6) and (b) m(t) = 1 when t < τ∗ (Section 5, Line 251), in my understanding, neither contextual path nor attention-shifting is used when t < min(τ∗, τ) . Will this cause incoherence?

---

> > > ### Author Response · Authors · 2023-08-18
> > > **Reply to Reviewer VS1k: Thanks for your response and further consideration!**
> > >
> > > Thanks for your response and for acknowledging the additional ablation study!
> > >
> > > > Is attention-shifting only applied in Video Generation Stage, but not applied in Interpolation Empowerment Stage, according to the caption of Figure 2?
> > >
> > > The attention shift is applied in both stages. In Figure 2, the denoising processes of both the Video Generation Stage and the Interpolation Empowerment Stage go through Diffusion U-Net, in which the proposed attention operation (Equation 6) at every denoising step is applied.
> > >
> > > > Considering (a) attention-shifting is not used when $t < τ$ (Equation 6) and (b) $m(t) = 1$ when $t < τ^∗$ (Section 5, Line 251), in my understanding, neither contextual path nor attention-shifting is used when $t < \min(τ^∗, τ)$. Will this cause incoherence?
> > >
> > > As the reviewer pointed out that when $t < \min(τ^∗, τ)$, neither ***contextual path*** nor ***attention to contextual contents*** is used. In our original rebuttal, the term “attention shift” refers to the overall attention operation (*not only the attention to contextual contents*) in Equation 6, and we apologize for the confusion caused by it.  We consider and experimentally showcase that this would not inherently cause contextual incoherence, but instead enhance semantic coherence:
> > >
> > > - Contextual coherence, such as the scene's general layout, shapes, and critical features, can be established during the denoising process's early stage (i.e., when $t$ is large).
> > > - When $t$ is small, paying attention to the contents within one frame conditioned on semantic information is necessary to ensure semantic coherence. In contrast, excessive attention to contextual contents would generate frames with minor differences, making videos into "trembling gifs".
> > >
> > > Therefore, we filter out the contextual content in both attention and interpolation during the denoising process’s late stage (when $t$ is small). Certainly, if we define $m(t)$ as a more elegant function varying through time steps, the performance of interpolation may be further enhanced. Still, the step function does not inherently lead to contextual incoherence.
> > >
> > > For attention strategy, the results of applying attention to contextual contents in the whole denoising process are shown in Section 5.2 (E). As for the interpolation module, we showcase the balance between two paths in Section 5.2 (Dual-path interpolation).

---

> > > > ### Comment · Reviewer_VS1k · 2023-08-19
> > > >
> > > > Thanks for the reply! From my experiment, I also found that excessive attention to contextual contents when t is small would generate frames with minor differences, which is not satisfying. However, I still have the concern that, disregarding contextual contents when t is small will cause fine-grained incoherence, such as the stamen in your demo gif (in supplementary material).
> > > >
> > > > As the authors have addressed most of my concerns, I would like to raise my score to borderline accept.

---

### Author Rebuttal · Authors · 2023-08-10

We thank all reviewers for engaging in the review process. Our code will be made public upon acceptance.

We are deeply encouraged by positive comments from the reviewers. We appreciate the recognition and endorsement of our proposed zero-shot pipeline, such as acknowledging its qualities as interesting (VS1k and 6FyB), novel (6FyB and yWMb), effective (yWMb), and technically sound (mgrm). VS1k, yWMb, and 6FyB agree that our method generates videos with high per-frame quality (fidelity). Both mgrm and 6FyB agree that our paper is “clear”, “well-written”, and “easy to understand”.

---

In our individual replies, we attempted to address specific questions and comments as clearly and detailed as possible.

Moreover, we added several additional results to the one-page PDF and the individual replies to strengthen our work overall. Here, we briefly summarize these additional experiments and evaluations:

- Supplemented ablation study of (a) numerical comparison on the test set (b) combining our techniques with Text2Video-Zero.
- Quantitative comparison of interpolation module.
- Quantitative results in an enlarged test set.
- Qualitative results of Tune-A-Video under complex conditions.
- Visualization of CLIP similarities on cases between all per-frame prompts in our test set.
- Qualitative results of using other LLMs.
- Qualitative results of videos with multiple subjects.
- Qualitative results of videos with straightforward motions.

---

We hope that these additional results further strengthen Free-Bloom’s position as the state-of-the-art generative model of zero-shot text-to-video generation and demonstrate:

- Our pipeline is novel to combine the LLM director and the LDM animator and is the first T2V work capable of generating semantic-coherent videos (such as the whole process of flower blooming rather than a set of “moving images”) in a zero-shot and training-free manner.
- Our Free-Bloom can generate flexible and versatile videos according to one unified prompt in a general level of scenes.
- Our novel techniques of sampling, attention, and interpolation can greatly improve the temporal coherence of the generated videos while being faithful to the rich semantic content.
- Our Free-Bloom is robust to different LLMs and is compatible with the existing LDM-based extensions.

---

### Decision · Program_Chairs · 2023-09-21

**Decision:**

Accept (poster)

**Comment:**

The paper received 5/7/5/7 ratings from reviewers. The initial review raised concerns regarding novelty/contribution of the paper as well as experimental evaluations. Those concerns were addressed through rebuttal discussion and two of the reviewers raised their scores. The AC acknowledges that the best practice of utilizing LLM and LDM still need exploration and this work benefit the research development over the direction. Taking all those into consideration, the AC recommends accepting the paper.